# Contextual Learning for Anomaly Detection in Tabular Data

**Spencer King** [*] [†]                                                           *sdk81722@uga.edu*
*School of Computing*
*University of Georgia, Athens, GA, USA*

**Zhilu Zhang** [*]                                                              *zhazhilu@amazon.com*
*Amazon Web Services, Seattle, WA, USA*

**Ruofan Yu** [*]                                                                 *ruofan@amazon.com*
*Amazon Web Services, Seattle, WA, USA*

**Baris Coskun**                                                                 *barisco@amazon.com*
*Amazon Web Services, Seattle, WA, USA*

**Wei Ding**                                                                      *dingwe@amazon.com*
*Amazon Web Services, Seattle, WA, USA*

**Qian Cui** [‡]                                                                  *cuiqia@amazon.com*
*Amazon Web Services, Seattle, WA, USA*

**Reviewed on OpenReview:** *https://openreview.net/forum?id=PmqZslRENW*

## Abstract

Anomaly detection is critical in domains such as cybersecurity and finance, especially when working with large-scale tabular data. Yet, unsupervised anomaly detection—where no labeled anomalies are available—remains challenging because traditional deep learning methods model a single global distribution, assuming all samples follow the same behavior. In contrast, real-world data often contain heterogeneous contexts (e.g., different users, accounts, or devices), where globally rare events may be normal within specific conditions. We introduce a *contextual learning framework* that explicitly models how normal behavior varies across contexts by learning conditional data distributions $P(\mathbf{Y} \mid \mathbf{C})$ rather than a global joint distribution $P(\mathbf{X})$. The framework encompasses (1) a probabilistic formulation for context-conditioned learning, (2) a principled bilevel optimization strategy for automatically selecting informative context features using early validation loss, and (3) theoretical grounding through variance decomposition and discriminative learning principles. We instantiate this framework using a novel conditional Wasserstein autoencoder as a simple yet effective model for tabular anomaly detection. Extensive experiments across eight benchmark datasets demonstrate that contextual learning consistently outperforms global approaches—even when the optimal context is not intuitively obvious—establishing a new foundation for anomaly detection in heterogeneous tabular data.

## 1 Introduction

Unsupervised anomaly detection (AD) is a critical field focused on identifying patterns that deviate from expected behavior, with applications spanning a wide range of tasks like security, fraud detection, and system reliability Chandola et al. (2009); Aggarwal (2017). The core challenge is not merely detecting new or unseen

---

[*]These authors contributed equally.
[†]Work done while interning at Amazon Web Services.
[‡]Corresponding Author.

objects, but discerning whether such observations align with the learned notion of "normal," or truly represent anomalies.

Recent advances in deep learning have driven a wave of neural network-based methods for unsupervised anomaly detection Landauer et al. (2023); Pang et al. (2021b). Despite methodological differences, most deep approaches model the joint distribution of all features, implicitly assuming that data patterns are homogeneous across conditions. In practice, patterns often vary significantly with context. For example, in cybersecurity, a spike in data transfer may be benign for a user who routinely performs backups but suspicious for a database administrator. Ignoring such context can degrade detection accuracy.

To address this challenge, we argue probabilistically and demonstrate empirically the effectiveness of modeling conditional distributions over joint distributions for anomaly detection. We advocate for a *contextual learning* framework, which models conditional distributions to capture intra-context regularities while isolating inter-context variability, thereby enhancing detection performance without requiring separate models for each condition.

Selecting an appropriate context feature is a critical aspect of contextual learning. It is often not obvious what feature one should condition on when modeling the data. Indeed, the choice of conditioning variable can significantly influence model performance. To address this, we develop a bilevel optimization methodology to automatically identify the context features based on early validation loss. Requiring only a small set of validation samples, our approach is simple, model-agnostic, and demonstrates consistent benefits across a wide range of datasets.

To demonstrate the framework's utility and empirically validate our claim, we build upon the Wasserstein autoencoder (WAE) Tolstikhin et al. (2019) to develop the conditional Wasserstein autoencoder (CWAE), a novel conditional variant designed to model conditional distributions for anomaly detection. We instantiate the framework using CWAE to conduct experiments using several tabular AD benchmark datasets containing inherent contextual information (e.g., users, accounts, or hosts). We demonstrate that incorporating contextual information with CWAE enables more robust anomaly detection across diverse and complex datasets, and offers significant advantages in improving detection accuracy.

Main contributions:

- We introduce a general contextual learning framework for anomaly detection.

- We provide a probabilistic formulation and theoretical justification grounded in variance decomposition and discriminative learning.

- We propose a bilevel optimization strategy for context selection using early validation loss.

- We instantiate the framework with a novel CWAE model and demonstrate consistent performance gains across diverse datasets.

- We demonstrate that conditional models in general can outperform both their non-contextual counterparts of the same architecture and state-of-the-art (SOTA) unconditional models.

## 2 Related Work

Anomaly detection has been a long-standing area of research, with early work dominated by statistical and distance-based approaches. Classical statistical methods such as Principal Component Analysis (PCA) Shyu et al. (2003) and distance-based techniques like k-Nearest Neighbors Liao & Vemuri (2002) and clustering-based methods Münz et al. (2007) aim to identify points that deviate from expected norms in the data distribution. While effective in lower dimensions, these techniques struggle in high-dimensional settings due to the "curse of dimensionality."

More recent advances leverage deep neural networks to learn expressive representations for anomaly detection in complex data. Notable lines of work include one-class classification with deep feature embeddings Ruff et al. (2018; 2021), autoencoder-based approaches that reconstruct normal samples Gong et al. (2019); Zhou &

Paffenroth (2017); Zhao et al. (2017); Liu et al. (2021), and generative models like GANs Schlegl et al. (2017); Sabuhi et al. (2021), normalizing flows Gudovskiy et al. (2022), and diffusion models Livernoche et al. (2023). However, several studies Kirichenko et al. (2020); Zhang et al. (2021) have shown that generative models trained to match the marginal data distribution can fail to identify out-of-distribution or context-specific anomalies, as they may assign high likelihood to atypical but structurally plausible inputs.

To address this, self-supervised learning methods Golan & El-Yaniv (2018); Wang et al. (2019b); Chen et al. (2020); Tack et al. (2020) recast anomaly detection as an auxiliary classification task, such as predicting data augmentations or transformations. While effective in image domains, extending such techniques to tabular or categorical data is non-trivial Bergman & Hoshen (2020); Qiu et al. (2021); Shenkar & Wolf (2022), and performance can degrade when the auxiliary task is misaligned with anomaly structure.

Orthogonal to these, subspace anomaly detection methods aim to discover subsets of features or projections where anomalies become more apparent. Classical examples include axis-parallel subspace approaches Kriegel et al. (2009), projection pursuit, and local subspace analysis Aggarwal & Yu (2013); Sathe & Aggarwal (2016). These methods, including randomized hashing Sathe & Aggarwal (2016), avoid modeling the full joint distribution by seeking dimensions that isolate outliers. However, they typically treat all features equally and do not distinguish between context and behavioral attributes.

In contrast, context-aware or conditional anomaly detection explicitly models how behavioral features deviate from expected values given a specific context Song et al. (2007). For instance, CADENCE Amin et al. (2019) models conditional probabilities for discrete events in high-cardinality logs, while the joint deep variational generative model of Shulman Shulman (2019) separates contextual and behavioral attributes for anomaly scoring. Similarly, UCAD Li et al. (2022) detects anomalous database operations by comparing operation semantics against contextual intent, and Moore & Morelli (2024) estimate context-conditioned distributions via normalizing flows for time series anomaly detection. These approaches highlight the benefits of modeling conditional distributions over global joint distributions, but they typically either require contextual and behavioral variables to be specified in advance or focus on more structured settings such as event records, logs, database operations, or time series.

Our work builds on this literature in three key ways. First, we target tabular data—a setting with less inherent structure and no predefined context—making the identification and modeling of contextual dependencies significantly more challenging. Second, unlike prior methods that rely on dual encoders or complex density estimators, we propose a lightweight yet effective CWAE that models $P(\text{content} \mid \text{context})$ through a simple reconstruction objective with MMD regularization. Third, we introduce a practical, unsupervised context selection strategy based on early validation loss, allowing the framework to automatically discover informative context features without supervision. Together, these components form a unified *contextual learning framework* for anomaly detection that is both conceptually simple and broadly applicable. This perspective reveals that meaningful contextual structure often exists even when not explicitly labeled, and that leveraging such structure can significantly enhance detection performance with minimal architectural complexity.

Additionally, our work is related to the extensive literature on statistical process control (SPC) and multivariate statistical process control (MSPC), which monitor deviations from learned baselines using control limits. Early univariate control charts date back to Shewhart Shewhart (1931), while multivariate extensions include Hotelling's $T^2$ charts Hotelling (1947) and Mahalanobis-distance-based detectors Mahalanobis (1936), as reviewed in standard SPC references Montgomery (2012). PCA-based MSPC further decomposes variability into latent subspaces and residual components, yielding statistics such as $T^2$ and squared prediction error (SPE) Jackson & Mudholkar (1979). More recently, autoencoder-based process monitoring methods replace linear subspace models with nonlinear representations, while using reconstruction error or latent-space monitoring statistics for fault detection Qian et al. (2022); Biegel et al. (2022).

Conceptually, these methods share the intuition that anomalies should be assessed relative to an appropriate baseline or operating regime. However, classical SPC and MSPC techniques typically rely on assumptions such as Gaussianity, linear structure, or known operating regimes, and are primarily developed for continuous process variables. In contrast, our framework generalizes this conditional monitoring perspective to high-dimensional, heterogeneous, and categorical tabular data without requiring explicit regime labels.

## 3 Preliminaries

Consider a random variable $\mathbf{X} \in \mathbb{R}^d$ representing data with $d$ features. We assume that $\mathbf{X}$ follows distribution $P(\mathbf{X})$. The anomaly detection problem aims to determine whether $\mathbf{X}$ is typical with respect to $P(\mathbf{X})$ or anomalous.

**Definition (Context-Conditional Anomaly Detection).** Let $\mathbf{X} = (\mathbf{C}, \mathbf{Y}) \in \mathbb{R}^k \times \mathbb{R}^{d-k}$ where $\mathbf{C}$ represents $k$-dimensional context features and $\mathbf{Y}$ represents $(d-k)$-dimensional content features. Given a training dataset $\mathcal{D} = \{(\mathbf{c}_i, \mathbf{y}_i)\}_{i=1}^n$ drawn from the distribution of normal events, our objective is to learn a score function

$$S : \mathbb{R}^k \times \mathbb{R}^{d-k} \to \mathbb{R}^+$$

that assigns an anomaly score $S(\mathbf{X})$ to each data point $\mathbf{X}$. Then based on this anomaly score, we set a threshold $\tau$ so that points with scores $S(\mathbf{X})$ exceeding $\tau$ are labeled as anomalies.

In traditional anomaly detection research, a common approach is to model the joint probability distribution over the entire feature space and to construct a single decision boundary to separate normal and anomalous behaviors. For instance, suppose $\mathbf{X} = \{A, B, C, D\}$ with four features, and we assume $\mathbf{X}$ follows a joint distribution $P(\mathbf{X}) = P(A, B, C, D)$. We estimate $P(\mathbf{X})$ from the training data and then derive an anomaly score directly from the learned distribution. Typically, observations with lower probability under $P(\mathbf{X})$ are considered more likely to be anomalous. Formally:

$$\text{Score}(\mathbf{X}) = S(P(\mathbf{X})),$$

or some other monotonic transformation of $P(\mathbf{X})$, i.e., $-\log P(\mathbf{X})$. To determine whether a new data point $\mathbf{X}$ is anomalous, the model compares the score against a predetermined threshold $\tau$:

$$\text{Label}(\mathbf{X}) = \begin{cases} \text{Normal} & \text{if Score}(\mathbf{X}) \leq \tau \\ \text{Anomalous} & \text{if Score}(\mathbf{X}) > \tau. \end{cases}$$

Here the single decision boundary $\tau$ is derived from training data, i.e., the maximum anomaly score.

A model using $P(\mathbf{X})$ for anomaly detection without conditional modeling relies on an implicit assumption of homogeneity across different contexts. However, this assumption may fail in practice, particularly when dealing with large-scale and complex datasets, as discussed in Section 1. If the context significantly influences the anomaly patterns, this assumption introduces noise into the learned boundary. For such scenarios, a single decision boundary derived from the joint distribution might not adequately capture the diverse anomaly patterns. Next we introduce a contextual conditioning approach that effectively isolates the variability across different contexts, enabling the model to learn more precise boundaries.

### 3.1 Contextual Learning Formulation

To address the heterogeneous nature of the problem, we hypothesize that a subset of features—referred to as **optimal context features**—captures the greatest diversity of anomaly patterns across their unique values, thereby defining distinct contexts. For simplicity, interpretability, and computational stability, we focus on a single feature that exhibits this maximal diversity, denoted as the **context feature**, while treating the remaining features as **content features** that characterize anomaly patterns within each context.

In practice, the partition between context and content features is entirely data-driven. Candidate context features are drawn from the observed feature set, and exactly one feature is selected as context via the validation-based procedure described in Section 5.3. All remaining features are treated as content and modeled conditionally. This partition is dataset-dependent and does not rely on prior semantic assumptions. When a candidate context is uninformative, the selection procedure naturally favors the no-context (global) baseline.

Focusing on a single feature offers several practical advantages. First, it enhances interpretability—practitioners can directly observe which variable defines the contextual partition. Second, it avoids the combinatorial explosion of context groups that arises under multi-feature conditioning, which can lead to

data sparsity and unstable estimation when certain feature combinations are rarely observed. Third, our empirical findings indicate that well-chosen single features often capture the dominant axis of contextual variation, while remaining dependencies can be effectively modeled through the content features. Importantly, this simplification does not restrict the generality of our framework: all theoretical results, including those on variance decomposition and discriminative learning advantages, naturally extend to the multi-feature case where $\mathbf{C} \in \mathbb{R}^k$ with $k > 1$. Future work may explore multi-feature context construction when single-feature conditioning proves insufficient.

As established in our definition above, we denote $\mathbf{C}$ as $k$-dimensional context features and $\mathbf{Y}$ as $(d-k)$-dimensional content features, where $k < d$. Given a training dataset $\mathcal{D} = \{(\mathbf{c}_i, \mathbf{y}_i)\}_{i=1}^n$ drawn from the distribution of normal events, our objective is to learn a score function

$$S : \mathbb{R}^k \times \mathbb{R}^{d-k} \to \mathbb{R}^+$$

where $S(\mathbf{c}, \mathbf{y})$ outputs an anomaly score for the context and content observation $(\mathbf{c}, \mathbf{y})$. Unlike the conventional approach that models $P(\mathbf{X})$, we now consider the conditional distribution $P(\mathbf{Y} \mid \mathbf{C})$. By focusing on $P(\mathbf{Y} \mid \mathbf{C} = \mathbf{c})$, we may define context-wise thresholds $\tau_{\mathbf{c}}$ for each $\mathbf{c} \in \text{supp}(\mathbf{C})$.

By leveraging this conditional structure, the model learns a separate decision boundary for each value $\mathbf{c}$ that $\mathbf{C}$ can take. In other words, it adapts the anomaly detection rule to the specific subspace partitioned by each context value. Thus, instead of relying on a single threshold $\tau$, we can define a context-dependent threshold $\tau_{\mathbf{c}}$.

The resulting context-conditioned anomaly detection rule becomes:

$$\text{Label}(\mathbf{c}, \mathbf{y}) = \begin{cases} \text{Normal}, & \text{if } S(\mathbf{c}, \mathbf{y}) \leq \tau_{\mathbf{c}}, \\ \text{Anomalous}, & \text{if } S(\mathbf{c}, \mathbf{y}) > \tau_{\mathbf{c}}. \end{cases}$$

This approach allows the model to more accurately profile the true boundaries of normal behavior by accounting for contextual variability, rather than relying on a single, global threshold. This formulation constitutes the core of our contextual learning framework.

## 4 On the Advantages of Contextual Learning

Conceptually, our contextual learning paradigm—defined as conditional modeling of content given context—parallels discriminative logic in the classic generative-discriminative framework. While generative methods strive to capture the joint distribution of inputs and labels (analogous to content and context in our framework), they can be error-prone if assumptions about the data-generating process are misspecified. By contrast, a conditional or discriminative approach sidesteps modeling the context distribution altogether, focusing solely on how content depends on context. This distinction is particularly salient when context variables (e.g., userIDs) have high cardinality, since modeling $P(\mathbf{C})$ or the full joint distribution $P(\mathbf{Y}, \mathbf{C})$ becomes prohibitively complex, risking a curse of dimensionality.

Such complexity motivates Vapnik's principle (Vapnik, 1995), which asserts that one should not solve a more general, and potentially more difficult, problem than necessary to meet the ultimate predictive goal. By focusing on $P(\mathbf{Y} \mid \mathbf{C})$, our method reduces the risk of model misspecification surrounding context variables and streamlines parameter estimation in high-dimensional spaces. In practice, this can lead to more robust and data efficient performance, which is an advantage also highlighted by extensive generative vs. discriminative comparisons in the classification literature. We illustrate this in more detail in the following subsections.

Furthermore, we illustrate the advantage of contextual learning from the perspective of variance reduction by the conditioned context variable. We argue that with an optimal choice of context variable, the conditional variance given each context would be smaller, resulting in less spread among data points, making deviations or anomalies more pronounced and easier to detect.

### 4.1 Model Misspecification Risks

Several theoretical and empirical studies substantiate these advantages for conditional or discriminative approaches. Ng & Jordan (2001) provides a rigorous comparison of Logistic Regression versus Naïve Bayes, demonstrating that although the generative model can dominate in very low-data regimes (due to stronger assumptions reducing variance), the discriminative model typically surpasses it once the sample size becomes sufficiently large. This phenomenon occurs precisely because a discriminative model directly optimizes the conditional likelihood pertinent to the task—akin to estimating $P(\mathbf{Y} \mid \mathbf{C})$ rather than the full joint distribution. The asymptotic analysis by (Liang & Jordan, 2008) further clarifies how misspecification in modeling $P(\mathbf{C})$ can degrade a generative model's performance, while a discriminative estimator that only captures $P(\mathbf{Y} \mid \mathbf{C})$ remains robust. Lasserre et al. (2006) shows how purely generative assumptions can degrade classification accuracy when they are violated, and how incorporating a discriminative component can improve robustness and prediction quality.

### 4.2 Curse of Dimensionality

In the high-dimensional context space, Hastie et al. (2009) discusses how focusing on the conditional distribution helps avoid unnecessary parameter explosion, while Bishop (2006) and Murphy (2012) provide extensive treatments of both paradigms, showing that discriminative techniques can be more robust to structural mismatches in the data generation process. These results generalize beyond simple supervised learning to broader scenarios such as one-class classification or semi-supervised training, where focusing on the part of the distribution necessary for the objective—be it anomaly detection or prediction—can reduce parameter complexity and improve predictive accuracy. Hence, in contexts where one can define a meaningful "background" subset of variables, instead of learning the full joint distribution, conditioning on them directly can provide both practical and theoretical advantages in model efficiency. By extension, in high-dimensional scenarios, focusing solely on $P(\mathbf{Y} \mid \mathbf{C})$ avoids the exponential blow-up in complexity that can come with modeling or sampling from $P(\mathbf{C})$. Consequently, conditional modeling can confer both practical and theoretical gains in efficiency and accuracy, making it a compelling strategy whenever a meaningful separation between context and content variables is possible.

### 4.3 Variance Reduction

In this section, we compare joint vs. conditional anomaly detection from a variance-reduction standpoint. Let

$$\mathbf{Y} \in \mathbb{R}^{d-k} \quad \text{(content features)}, \qquad \mathbf{C} \in \mathbb{R}^k \quad \text{(context features)}.$$

A *joint* model learns the full density $P(\mathbf{Y}, \mathbf{C})$, however, the *marginal*

$$P(\mathbf{Y}) = \int P(\mathbf{Y}, \mathbf{C}) \, \mathrm{d}\mathbf{C} = \int P(\mathbf{Y} \mid \mathbf{C}) \, P(\mathbf{C}) \, \mathrm{d}\mathbf{C}$$

mixes together data from all contexts.

By contrast, a *conditional* approach directly learns $P(\mathbf{Y} \mid \mathbf{C})$ and scores each sample within its own context slice. The key theoretical insight is the law of total variance:

$$\mathrm{Var}(\mathbf{Y}) = \underbrace{\mathbb{E}_{\mathbf{C}}\big[\mathrm{Var}(\mathbf{Y} \mid \mathbf{C})\big]}_{\substack{\text{average "within-context"} \\ \text{variance}}} + \underbrace{\mathrm{Var}_{\mathbf{C}}\big(\mathbb{E}[\mathbf{Y} \mid \mathbf{C}]\big)}_{\substack{\text{"between-context"} \\ \text{variance}}}.$$

A joint detector must accommodate both terms, whereas a context-conditional detector only needs to cope with $\mathrm{Var}(\mathbf{Y} \mid \mathbf{C} = \mathbf{c})$, which is typically much smaller than the total $\mathrm{Var}(\mathbf{Y})$.

So ideally if we can find contexts to partition the data so that

$$\mathrm{Var}(\mathbf{Y} \mid \mathbf{C} = \mathbf{c}) \ll \mathrm{Var}(\mathbf{Y})$$

holds for most $\mathbf{c} \in \mathrm{supp}(\mathbf{C})$, conditioning makes our anomaly detectors both sharper and more sensitive to small deviations from normal, context-specific patterns. For reconstruction based models, this variance reduction yields the following practical benefits:

1. **Higher signal-to-noise ratio.** Normal samples concentrate more tightly around their reconstructions, making anomalous deviations stand out. We can choose a threshold $\tau_{\mathbf{c}}$ for each context $\mathbf{c}$ based on the smaller $\mathrm{Var}(\mathbf{Y} \mid \mathbf{C} = \mathbf{c})$, instead of a single conservative $\tau$ for the full marginal distribution.

2. **More stable and efficient learning.** Each conditional model only needs to capture a homogeneous slice of the data, leading to more stable gradient updates, faster convergence, and fewer mixed-mode reconstruction errors, as the model doesn't need to accommodate conflicting patterns from different contexts simultaneously.

We now formalize the above intuition with a heuristic concentration-based argument linking conditional variance reduction to anomaly-score separability.

**Proposition 1** (Heuristic link between conditional variance reduction and anomaly score separability)**.** *Let $S(Y)$ denote an anomaly score derived from a reconstruction-based model. Assume that for normal data and any fixed context $c$, the conditional score distribution $S(Y) \mid C = c$ is sub-Gaussian with variance proxy $\sigma_c^2$, where lower conditional variability in $Y \mid C = c$ induces a smaller variance proxy for $S(Y) \mid C = c$. Then, for anomalous samples whose reconstruction scores remain separated from those of normal samples by a fixed margin, conditioning on $C$ can improve anomaly-score separability relative to scoring under the marginal distribution $P(Y)$.*

*Proof sketch.* Under the sub-Gaussian assumption, the tail probability of the reconstruction score for normal samples satisfies

$$\mathbb{P}(S(Y) > \tau \mid C = c) \leq \exp\left(-\frac{(\tau - \mu_c)^2}{2\sigma_c^2}\right),$$

where $\mu_c = \mathbb{E}[S(Y) \mid C = c]$ and $\sigma_c^2$ is the corresponding variance proxy. By the law of total variance,

$$\mathrm{Var}(Y) = \mathbb{E}_C[\mathrm{Var}(Y \mid C)] + \mathrm{Var}_C(\mathbb{E}[Y \mid C]),$$

so marginal scoring under $P(Y)$ mixes within-context variability with additional between-context variability. Heuristically, this can induce a broader normal score distribution than context-conditional scoring, yielding looser concentration. Consequently, for a fixed false-positive rate, the context-dependent threshold $\tau_c$ can be tighter than the corresponding marginal threshold, thereby improving the potential separation between normal and anomalous score distributions. $\qquad\square$

## 5 Instantiating Contextual Learning with CWAE

To demonstrate the contextual learning framework in practice, we instantiate it using the **C**ontextual **W**asserstein **A**uto-**E**ncoder (CWAE). CWAE is a novel, generative, and reconstruction-based model designed to be straightforward and interpretable, operationalizing the framework by modeling context-conditional distributions for anomaly detection. While CWAE achieves strong performance, its primary purpose is to illustrate the effectiveness and generality of the contextual learning framework rather than to serve as the main contribution of this work. Comprehensive implementation details are provided in the Appendix A.

### 5.1 CWAE Model Structure

The CWAE model is designed to learn the conditional distribution of input features given contextual features. This section outlines the objective, architecture, and loss function.

#### 5.1.1 Model Objective

CWAE aims to conditionally reconstruct input content features based on associated context features. By learning a latent representation that combines content and context, the model identifies anomalies through reconstruction loss, where higher reconstruction loss indicates a greater likelihood of anomalous behavior. This framework allows CWAE to adapt to varying contextual conditions, which is critical for datasets with complex or diverse anomaly patterns.

### 5.1.2 Model Architecture

As displayed in Figure 1, CWAE consists of three main components: embedding layers, an encoder, and a decoder.

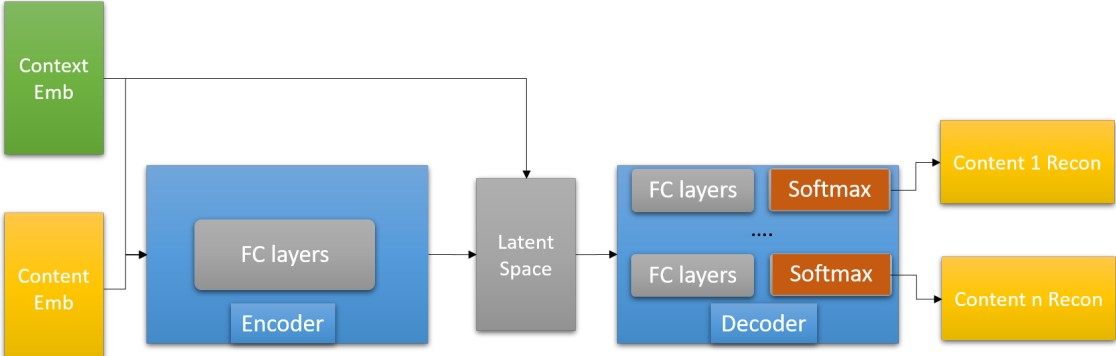

Figure 1: Architecture of the CWAE model utilized in the experiments.

**Embedding Layers.** Two sets of embedding layers are employed—one for context features and another for content features. Each embedding layer is randomly initialized, with the first dimension corresponding to the number of unique feature values plus one (to handle unseen values) and the second dimension representing the embedding vector size.

**Encoder.** The encoder takes as input the concatenated embeddings of context and content features. This input is processed through fully connected (FC) layers to extract a latent representation. To retain contextual information, the output of these layers is concatenated with the original context embeddings to form the final latent representation.

**Decoder.** The decoder reconstructs the content features from the latent representation. A series of FC layers are used to map the latent representation back to the original feature space.

### 5.1.3 Loss Function

The loss function combines reconstruction accuracy with regularization to ensure effective learning. Formally:

$$\mathcal{L}_{\text{cwae}} = \sum_{i=1}^{d} \text{CE}(\mathbf{Y}_i, \widehat{\mathbf{Y}}_i)$$
$$+ \lambda \cdot \text{MMD}\Big(P(z), Q(z \mid X = \mathbf{C}, \mathbf{Y})\Big), \tag{1}$$

where $\mathbf{C}$ and $\mathbf{Y}$ represent context and content features, respectively; $\text{CE}(\mathbf{Y}_i, \widehat{\mathbf{Y}}_i)$ represents the cross-entropy reconstruction loss for content features and d is the dimension of content $\mathbf{Y}$; $\lambda$ is a weighting hyperparameter for the MMD term; $\text{MMD}\left(P(z), Q(z \mid X)\right)$ quantifies the maximum mean discrepancy between the latent distribution and a predefined prior (e.g., Gaussian). The reconstruction loss ensures that input features are accurately reconstructed, while the MMD regularization term aligns the latent distribution with the desired prior. The combined loss facilitates effective training and ensures the model captures meaningful contextual patterns.

### 5.2 Training and Inference

**Training.** During training, tabular features are first mapped to indices, resulting in integer vectors as input. These indices are passed through embedding layers, which are updated during training. The encoder

and decoder work together to reconstruct the content features, and the total loss is optimized to minimize reconstruction error and regularization discrepancy.

**Inference.** During inference, anomaly detection is performed by calculating the reconstruction loss for each instance. Samples with reconstruction loss exceeding the context-specific threshold (as defined in Section 6.1.3) are classified as anomalies. This approach allows the model to adapt to varying contextual conditions while maintaining robust detection performance.

### 5.3 Context Feature Selection

Selecting an appropriate context feature is one of the most critical and challenging aspects of the contextual learning framework. The chosen context fundamentally shapes how the model interprets patterns in the data and can have a substantial impact on detection performance. As shown in Tables 3, 4, and 5, contextual models consistently outperform their non-contextual counterparts when the context is well chosen, underscoring its central role in effective anomaly detection. Achieving strong results therefore requires a principled and effective strategy for selecting a context feature that is likely to yield good performance. This procedure operationalizes the context–content partition introduced in Section 3.1 by automatically selecting one observed feature as context and treating the remainder as content.

We propose to frame context selection as a bilevel optimization problem Sinha et al. (2018), where the goal is to optimize for the joint data likelihood $P(\mathbf{Y}, \mathbf{C}) = P(\mathbf{Y} \mid \mathbf{C}) \, P(\mathbf{C})$. In this setting, both the model parameters and the context feature are treated as variables to optimize. The inner loop learns the model weights for a given context, while the outer loop searches for the context feature that yields the best generalization performance.

To enable reliable context selection, our approach uses a small validation subset drawn from data that is *assumed to be predominantly normal*. This assumption is aligned with common settings in unsupervised and semi-supervised anomaly detection, where training data typically consist primarily or exclusively of normal samples Tax & Duin (2004); Ruff et al. (2018); Bergman & Hoshen (2020). It is also consistent with recent work that explicitly studies model selection and validation without labeled anomalies, including settings that use only normal data during both training and validation Cui et al. (2023) or rely on a small support set of normal samples to construct validation procedures in the absence of labeled validation data Fung et al. (2024).

This requirement is minimal, since only enough samples are needed to compute stable loss estimates rather than labeled anomalies. The validation data is used exclusively to evaluate generalization performance for context selection and is not used to train the anomaly detector itself, ensuring that the framework remains fully unsupervised with respect to anomaly labels. Such assumptions are common in practical anomaly detection deployments, where a short baseline period or trusted historical data is often available (e.g., during initial system deployment). When a clean validation subset is unavailable, practitioners may instead reserve a small portion of the training data as a proxy validation set, although this may slightly reduce selection reliability.

We note that robustness to significant anomaly contamination in unsupervised data constitutes a distinct and well-studied problem setting, often framed under contaminated data, noisy supervision, or weakly supervised anomaly detection. Recent surveys explicitly identify this as a separate line of research Jiang et al. (2023), and several methodological works propose dedicated mechanisms—such as iterative reweighting, pseudo-labeling, or contamination-aware objectives—to handle high contamination ratios Kim et al. (2023); Wu et al. (2022). Our work addresses a different setting and follows the conventional assumption that training and validation data are mostly benign. That said, because our bilevel selection procedure depends primarily on the *relative ranking* of candidate context features rather than absolute validation loss values, small amounts of contamination (e.g., 5–10%) are unlikely to fundamentally alter context rankings.

Naively training each conditional model to convergence is thorough, but can be prohibitively expensive. In practice, we only train the model for one epoch, and evaluate the model with validation loss afterwards. As a result, the computational cost of context selection scales linearly with the number of candidate context features, requiring one partial (single-epoch) training run per feature. In practice, this cost is modest relative

to full model training and is trivially parallelizable across candidate contexts. Consequently, the procedure is well-suited to modern multi-core or distributed training environments.

Our single-epoch evaluation strategy is motivated by extensive empirical and theoretical evidence showing that early training dynamics are strongly predictive of final generalization performance. Numerous prior works have successfully leveraged early-trained models for neural architecture search Liu et al. (2019); Bender et al. (2018), model pruning You et al. (2020), dataset pruning Paul et al. (2021), and learning with noisy labels Liu et al. (2020); Zhang & Sabuncu (2018). Studies have also shown that the first few epochs establish key characteristics of the learned representation Achille et al. (2017); Zhang et al. (2016); Frankle et al. (2020). In our context selection setting, the goal is not absolute performance but rather the relative ranking among context features, a signal that emerges early in training. Figure 3 supports this empirically, as the early-epoch proxy achieves better performance on five of eight datasets.

To ensure consistency across models with different probabilistic factorizations—e.g., $P(\mathbf{Y} \mid \mathbf{C})$ vs. $P(\mathbf{Y}, \mathbf{C})$—we evaluate all models using their joint log-likelihood. For conditional models, we apply the identity $P(\mathbf{Y}, \mathbf{C}) = P(\mathbf{Y} \mid \mathbf{C}) \, P(\mathbf{C})$, where $P(\mathbf{C})$ is estimated empirically from the training data. This formulation also penalizes rare context values via the $\log P(\mathbf{C})$ term, favoring parsimonious and meaningful conditioning. As motivated in Section 3.1, we only condition on one feature at a time as the context feature. As such, $P(\mathbf{C})$ can be easily obtained from the training data.

Putting everything together we present the proposed procedure for selecting context features in Algorithm 1. This data-driven selection method avoids heuristic filtering and provides a principled mechanism for identifying informative context features within the contextual learning framework.

---

**Algorithm 1** Context Selection via Validation Loss Minimization

---

Given candidate context features $\mathcal{K}$, training set $\mathcal{D}_{\mathrm{train}}$, validation set $\mathcal{D}_{\mathrm{val}}$ assumed to be predominantly normal, and model $\mathcal{M}$:

1. **Outer Loop (Context Optimization):**
   - For each $\mathbf{C} \in \mathcal{K}$:
     a. Initialize model $\mathcal{M}$ conditioned on context $\mathbf{C}$
     b. **Inner Loop (Model Training):** Train $\mathcal{M}$ on $\mathcal{D}_{\mathrm{train}}$ to optimize weights $w$ for 1 epoch
     c. Compute joint validation loss $\mathcal{L}_{\mathrm{val}} = -\log P(\mathbf{Y}, \mathbf{C})$ on $\mathcal{D}_{\mathrm{val}}$
2. Select context $\mathbf{C}^*$ with minimal validation loss

Return final context $\mathbf{C}^*$ and trained model $\mathcal{M}(\mathbf{Y} \mid \mathbf{C}^*)$

---

Figure 2 illustrates this process on the Census dataset using CWAE. We show the validation loss curves for the top and bottom three candidate context features, based on performance after a single epoch. The feature *detailed_occupation_recode* results in the lowest loss and is therefore selected.

Using this selection method, we achieve performance equal to or better than the non-contextual baseline in all but two datasets. Figure 3 shows the improvement in AUCROC for CWAE when using the selected context feature, compared to its non-contextual counterpart. In cases where the no-context option yields the best validation loss, it is retained as the final choice (marked in orange). Final context selections are summarized in Table 3 under the *Context* column.

This selection cost is comparable in spirit to commonly used procedures in anomaly detection and representation learning, such as feature screening, hyperparameter search, or early-stopping-based model selection, all of which require multiple partial training runs. While our approach incurs higher upfront cost than lightweight methods such as Isolation Forest, it remains practical for high-cardinality, heterogeneous tabular settings, where improved detection accuracy is often prioritized over minimal training time.

## 5.4 Advantages of the Wasserstein Approach

As outlined in Section 4, effective conditional modeling of $P(\mathbf{Y} \mid \mathbf{C})$ benefits from architectures that are both stable and efficient at inference. Reconstruction-based anomaly detection models—including standard

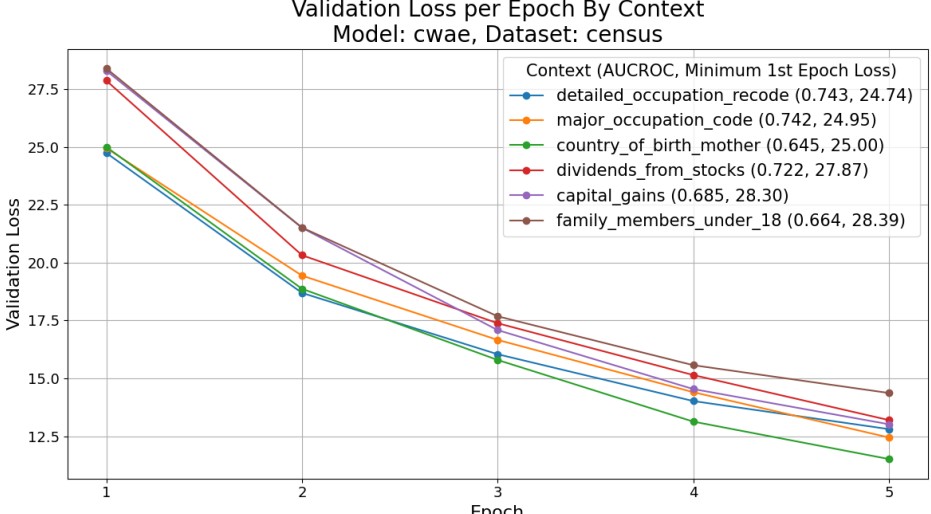

Figure 2: Validation loss curves for CWAE on the Census dataset, conditioned on different context features.

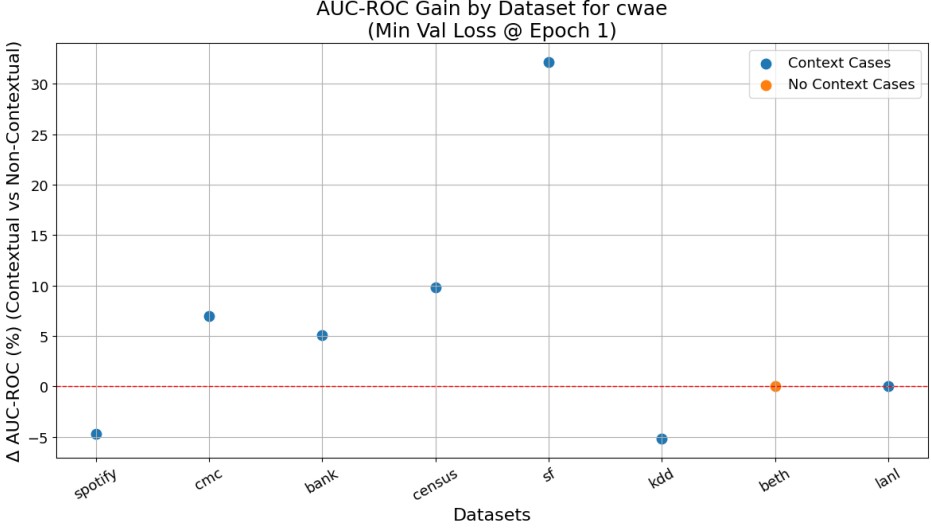

Figure 3: AUCROC improvement achieved by applying Algorithm 1 using only one epoch of training to select context for CWAE across datasets. Results show that one epoch validation loss provides a sufficiently reliable relative ranking signal for context selection on the majority of datasets. When early signals are uninformative, the method degrades gracefully toward the no-context baseline.

autoencoders Zhou & Paffenroth (2017); Gong et al. (2019), GAN-based approaches Schlegl et al. (2017), and variational autoencoders (VAEs) and their conditional variants Kingma & Welling (2014); Sohn et al. (2015)—have been widely used for modeling normal data distributions. However, stochastic latent-variable models such as VAEs often benefit from averaging over multiple latent samples at inference to obtain more stable anomaly scores, at the cost of additional computation.

Wasserstein Autoencoders (WAEs) Tolstikhin et al. (2019) replace the KL-based regularization used in VAEs with an optimal transport-based objective, which can be implemented using Maximum Mean Discrepancy (MMD) Gretton et al. (2012). This formulation supports deterministic encoders while regularizing the aggregated posterior to match a chosen prior. By avoiding Monte Carlo sampling at inference, deterministic

encoding can make anomaly scoring faster and more stable, aligning with the robustness and efficiency goals discussed in Section 4.

In this work, we instantiate CWAE with a deterministic encoder–decoder pair. This design choice reduces inference latency, improves score consistency, and maintains a well-regularized latent space, providing practical benefits over the Conditional VAE (CVAE) baseline while preserving the ability to model complex conditional distributions.

Importantly, these properties are especially well aligned with the goals of contextual anomaly detection. Because anomaly scores are computed and compared across heterogeneous contexts, score stability and low variance at inference time are critical: stochastic latent sampling can introduce unnecessary noise that obscures true context-dependent deviations. Deterministic encoding ensures that anomaly scores reflect structural differences in the data rather than sampling variability, leading to more consistent context-wise thresholds and more reliable cross-context comparisons. We emphasize that this design choice is complementary to the broader contextual learning framework rather than essential to its validity; alternative conditional models can be substituted within the framework, but the Wasserstein formulation provides a particularly stable and practical instantiation.

## 6 Experiments

### 6.1 Experimental Setup

#### 6.1.1 Dataset Selection

To evaluate the effectiveness of the contextual learning framework, we instantiate it using CWAE and assess its performance on selected datasets spanning diverse domains, including finance, cybersecurity, demographics, and network intrusion detection. Each dataset is designed to facilitate the study of anomaly detection with contextual learning, where anomalies are defined by specific criteria such as customer behavior, malicious events, or demographic thresholds.

We started with datasets introduced in the AD literature, such as those discussed in Han et al. (2022), and expanded our selection to include additional datasets originally used for multi-class classification. Collectively, we have compiled a suite of eight datasets for comprehensive empirical evaluation, which includes the Bank Marketing dataset (bank) Moro et al. (2012), the Beth Cybersecurity dataset (beth) Highnam et al. (2021), the Census dataset (census) cen (2000), the CMC dataset (cmc) Lim (1999), the KDD dataset (kdd) kdd (1999), the LANL dataset (lanl) Turcotte et al. (2019), the Solar Flares dataset (sf) sol (1989) and the Spotify dataset (spotify) Choksi (2021). Details about the datasets can be found in Table 1. These datasets vary significantly in size, feature count, and anomaly prevalence, providing a comprehensive foundation for robust evaluation. Complete implementation and evaluation details to facilitate reproducibility are provided in the Appendix A.

#### 6.1.2 Data Complexity Ranking

Understanding dataset complexity is essential for ensuring that contextual learning methods are evaluated across a broad spectrum of dataset difficulties. By considering complexity, we can better assess how contextual learning behaves in both simpler and more challenging anomaly detection scenarios. This perspective allows us to examine whether the benefits of contextual learning hold consistently across varying levels of difficulty. To capture these complexities, we adopt several metrics from AD literature Pang et al. (2021a), tailored to reflect the specific characteristics of our datasets. It is important to **note that the metrics are used solely as post-hoc diagnostic descriptors to contextualize empirical results**, rather than as evaluation criteria, optimization objectives, or the basis for any methodological claims. These metrics are defined as:

- **Value Coupling Complexity** ($K_{vcc}$): Quantifies the impact of similarity among outliers (homophily) on detection effectiveness. Higher $K_{vcc}$ values indicate greater complexity, as coupled outliers introduce noise and correlations that hinder detection.

- **Heterogeneity of Categorical Distribution** ($K_{\text{het}}$): Measures the diversity in the frequency distribution of dominant categories across features. A higher $K_{\text{het}}$ value signifies increased difficulty in identifying outliers due to the varied distribution of categorical modes.

- **Outlier Inseparability** ($K_{\text{ins}}$): Represents the difficulty of separating outliers from normal data points. It ranks features based on the inverse frequency of their values, with higher $K_{\text{ins}}$ values suggesting that anomalies are less distinguishable from normal objects.

- **Feature Noise Level** ($K_{\text{fnl}}$): Captures the proportion of features where noise causes outliers to exhibit frequencies similar to or higher than normal data points. Higher $K_{\text{fnl}}$ values indicate increased detection difficulty due to noise obscuring distinctions.

Table 2 provides a comprehensive breakdown of dataset complexity across the four distinct metrics: $K_{\text{vcc}}$, $K_{\text{het}}$, $K_{\text{ins}}$, and $K_{\text{fnl}}$. For each dataset, we report the raw metric value, its min-max scaled version, and the corresponding rank. The overall complexity score (*Avg Scaled*) is computed as the average of the scaled scores across all four metrics. This aggregate score serves as a unified measure of dataset difficulty. The final ranking reflects this overall complexity, where higher scores correspond to more challenging datasets for anomaly detection, and lower rank values indicate relatively easier datasets.

Comparing Tables 1 and 2 shows that our complexity metrics capture challenges that basic statistics miss. For instance, *spotify* emerges as the most complex dataset despite appearing ordinary in size, feature count, and cardinality, highlighting the role of structural and contextual patterns in driving difficulty.

| Dataset Name | Number of Features | Size | Number of Anomalies | Anomaly Ratio | Avg Cardinality |
|---|---|---|---|---|---|
| bank | 11 | 41,188 | 4,640 | 11.27% | 5 |
| beth | 11 | 1,141,078 | 158,432 | 13.88% | 35,154 |
| census | 38 | 299,285 | 18,568 | 6.20% | 17 |
| cmc | 8 | 1,473 | 29 | 1.97% | 3.13 |
| kdd | 7 | 1,014,535 | 440 | 4.51% | 12 |
| lanl | 16 | 2,542,727 | 5,971 | 0.23% | 3,191 |
| sf | 11 | 1,066 | 43 | 4.03% | 3.73 |
| spotify | 17 | 113,550 | 2,412 | 2.12% | 4,615 |

Table 1: Overview of datasets with descriptive statistics.

| | $K_{vcc}$ | | | $K_{het}$ | | | $K_{ins}$ | | | $K_{fnl}$ | | | Overall Complexity | |
|---|---|---|---|---|---|---|---|---|---|---|---|---|---|---|
| Dataset | Score | Scaled | Rank | Score | Scaled | Rank | Score | Scaled | Rank | Score | Scaled | Rank | Avg Scaled | Rank |
| bank | 0.210 | 0.196 | 5 | 2.015 | 0.016 | 4 | 0.343 | 0.775 | 3 | 1.000 | 1.000 | 1 | 0.497 | 2 |
| beth | 0.916 | 1.000 | 1 | 2.929 | 0.037 | 3 | 0.016 | 0.032 | 7 | 0.179 | 0.179 | 5 | 0.312 | 4 |
| census | 0.450 | 0.469 | 3 | 3.500 | 0.049 | 2 | 0.238 | 0.536 | 4 | 0.286 | 0.286 | 4 | 0.335 | 3 |
| cmc | 0.038 | 0.000 | 8 | 1.579 | 0.007 | 6 | 0.348 | 0.786 | 2 | 0.000 | 0.000 | 8 | 0.198 | 7 |
| kdd | 0.055 | 0.019 | 7 | 1.278 | 0.000 | 8 | 0.159 | 0.357 | 6 | 0.500 | 0.500 | 2 | 0.219 | 6 |
| lanl | 0.369 | 0.377 | 4 | 1.939 | 0.015 | 5 | 0.002 | 0.000 | 8 | 0.009 | 0.009 | 7 | 0.100 | 8 |
| sf | 0.124 | 0.098 | 6 | 1.564 | 0.006 | 7 | 0.176 | 0.395 | 5 | 0.500 | 0.500 | 2 | 0.250 | 5 |
| spotify | 0.500 | 0.526 | 2 | 46.264 | 1.000 | 1 | 0.442 | 1.000 | 1 | 0.123 | 0.123 | 6 | 0.662 | 1 |

Table 2: Dataset complexity metrics with raw scores (*Score*), scaled scores (*Scaled*), and ranks (*Rank*) reported for each. Higher scores indicate greater complexity, while lower ranks correspond to higher complexity.

Conversely, datasets with large size or high cardinality, such as *beth* and *lanl*, rank lower in complexity, suggesting that these properties alone do not necessarily make anomaly detection harder. Overall, the complexity metrics offer a more nuanced basis for evaluating contextual learning across datasets of varying difficulty.

### 6.1.3 Contextual Thresholding for AUCROC Calculation

Traditional AUCROC calculation typically involves grid searching over a single, global decision threshold. In the no-context setting, this is straightforward since all samples share the same threshold. However, in

contextual learning, using a single global threshold can be suboptimal because score distributions may vary significantly across context groups. To address this, we adopt a per-context thresholding approach while ensuring that resulting AUCROC values remain directly comparable to the baseline.

Given a training set with a particular context feature selected, we define the maximum training loss within each context group as its contextual threshold $H_{\mathbf{C}_i}$. For the test set, each sample's anomaly score is normalized by its corresponding contextual threshold, producing a contextual ratio $R_{\mathbf{C}_i} = \frac{\text{anomaly score}}{H_{\mathbf{C}_i}}$. This normalization rescales scores so that they are on the same relative scale across different contexts, effectively reducing the per-group thresholding problem to a single normalized thresholding problem.

We then compute the AUCROC by grid searching over 100 evenly spaced thresholds across the range of $R_{\mathbf{C}_i}$. For example, step $k$ corresponds to a threshold of $0.01 \cdot k \cdot \max(R_{\mathbf{C}_i})$. Since both the contextual and no-context settings ultimately apply thresholding to a single unified score distribution (global anomaly scores in the baseline, normalized contextual ratios in the contextual case), the resulting AUCROC values are directly comparable across all model configurations.

### 6.1.4   Handling Unseen Contexts

Context features are embedded using lookup tables with an additional fallback index reserved for unseen values. During inference, any context value not observed during training is mapped to this fallback embedding. In these cases, context information is effectively unavailable, and anomaly scoring relies primarily on the learned content representation. This yields a well-defined fallback behavior that is comparable to a non-contextual baseline, without requiring extrapolation to unseen context embeddings.

Our evaluation pipeline distinguishes between samples whose context values were observed during training and those containing previously unseen context values. Unseen contexts are evaluated using the fallback embedding described above, allowing performance in this regime to be analyzed separately rather than being conflated with seen-context results. Implications for low-support and rare contexts are discussed further in Section 9.

### 6.1.5   Benchmark Models

To effectively benchmark our proposed model, we selected six SOTA anomaly detection models from the DeepOD package Xu: DSVDD Ruff et al. (2018), RDP Wang et al. (2019a), RCA Liu et al. (2021), ICL Shenkar & Wolf (2022), DIF Xu et al. (2023a), and SLAD Xu et al. (2023b). Based on our research, these models represent leading techniques in the anomaly detection literature. The DeepOD package provides comprehensive implementations of these methods within a unified testbed, enabling consistent evaluation. Each model was tested using the default hyperparameter settings provided by DeepOD, ensuring comparability across experiments. All models were implemented without incorporating contextual considerations for anomaly detection.

In addition to the DeepOD-based models, we include the DTE Livernoche et al. (2023) model, implemented from the available paper details. As one of the most recent diffusion-based approaches to anomaly detection, DTE introduces a fundamentally different architecture that broadens the diversity of our baseline set. We also evaluate the WAE Tolstikhin et al. (2019) model as a non-contextual baseline. WAE shares the same architecture as our proposed CWAE but treats all features uniformly as content, without distinguishing contextual variables, making it a strong reference point for isolating the effects of contextual modeling.

## 6.2   Performance Evaluation

In this section, we present the CWAE results obtained using the selection method described in Section 5.3. Tables 3 and 4 report the corresponding AUCROC scores and their rankings, respectively.

**CWAE achieves top overall performance.**   We evaluate model performance using two metrics: AUCROC and average rank. Ranks are computed per dataset, then averaged across all datasets to offer a balanced view that reduces the influence of outliers. CWAE attains the best overall performance, with the highest average AUCROC (0.797) and the lowest average rank (2.75) among all models.

| Dataset | Scaled Complexity | DSVDD | RDP | RCA | ICL | DIF | SLAD | DTE | WAE | CWAE | Context |
|---------|-------------------|-------|-----|-----|-----|-----|------|-----|-----|------|---------|
| bank | 0.497 | 0.455 | 0.582 | 0.649 | 0.519 | 0.580 | 0.464 | 0.582 | 0.654 | **0.687** | loan |
| beth | 0.312 | 0.895 | 0.998 | 0.997 | 0.953 | 0.988 | 0.995 | **0.999** | 0.996 | 0.996 | no_ctx |
| census | 0.335 | 0.443 | 0.629 | 0.701 | 0.637 | 0.669 | 0.673 | 0.410 | 0.676 | **0.743** | detailed_occupation_recode |
| cmc | 0.198 | 0.610 | 0.566 | **0.760** | 0.530 | 0.675 | 0.713 | 0.699 | 0.702 | 0.751 | Husbands_education |
| kdd | 0.219 | 0.615 | 0.924 | 0.880 | 0.501 | 0.928 | **0.939** | 0.883 | 0.674 | 0.639 | is_guest_login |
| lanl | 0.100 | **1.000** | 0.948 | 0.867 | **1.000** | 0.914 | 0.978 | **1.000** | 0.999 | 0.999 | SubjectLogonID |
| sf | 0.250 | 0.579 | 0.811 | 0.800 | 0.731 | 0.827 | 0.267 | 0.731 | 0.678 | **0.895** | C-class_flares_production_by_this_region |
| spotify | 0.662 | 0.403 | 0.537 | 0.501 | 0.585 | 0.480 | 0.531 | 0.431 | **0.697** | 0.665 | loudness |
| AVG | − | 0.625 | 0.749 | 0.769 | 0.682 | 0.758 | 0.695 | 0.717 | 0.760 | **0.797** | − |

Table 3: AUCROC scores, scaled complexity scores, and the selected context feature for all models across datasets. Bold and underlined values indicate the best AUCROC scores.

| Dataset | DSVDD | RDP | RCA | ICL | DIF | SLAD | DTE | WAE | CWAE |
|---------|-------|-----|-----|-----|-----|------|-----|-----|------|
| bank | 9 | 4 | 3 | 7 | 6 | 8 | 4 | 2 | **1** |
| beth | 9 | 2 | 3 | 8 | 7 | 6 | **1** | 4 | 4 |
| census | 8 | 7 | 2 | 6 | 5 | 4 | 9 | 3 | **1** |
| cmc | 7 | 8 | **1** | 9 | 6 | 3 | 5 | 4 | 2 |
| kdd | 8 | 3 | 5 | 9 | 2 | **1** | 4 | 6 | 7 |
| lanl | **1** | 7 | 9 | **1** | 8 | 6 | **1** | 4 | 4 |
| sf | 8 | 3 | 4 | 5 | 2 | 9 | 5 | 7 | **1** |
| spotify | 9 | 4 | 6 | 3 | 7 | 5 | 8 | **1** | 2 |
| AVG | 7.375 | 4.75 | 4.125 | 6 | 5.375 | 5.25 | 4.625 | 3.875 | **2.75** |

Table 4: Model ranks for all datasets. Bold and underlined values indicate the best (lowest) rank.

**Contextual learning improves over non-contextual baselines.** CWAE consistently outperforms its non-contextual counterpart WAE, which shares the same architecture, loss function, and training procedure but omits contextual conditioning. This comparison isolates the effect of conditioning itself, independent of how the context feature is chosen. Across all eight datasets, CWAE improves the average AUCROC from 0.760 (WAE) to 0.797, thereby demonstrating the benefit of conditional modeling per se. Compared to other leading non-contextual baselines of different architectures—RCA (0.769, 4.125), DIF (0.758, 5.375), RDP (0.749, 4.75), and DTE (0.717, 4.625)—CWAE remains the strongest overall.

**CWAE excels across datasets of varying complexity.** A dataset-wise breakdown reveals CWAE's flexibility and robustness:

- **spotify (high complexity)**: WAE achieves the best AUCROC (0.697), but CWAE is competitive at 0.665 and outperforms several baselines, including ICL (0.585) and RDP (0.537). This illustrates contextual modeling's effectiveness in challenging data regimes.

- **bank and census (moderate complexity)**: CWAE attains the top AUCROC on both datasets (0.687 and 0.743), surpassing RCA and WAE. This shows strong generalization to structured real-world tabular data.

- **lanl (low complexity)**: CWAE attains a near-perfect AUCROC of 0.999, just behind DSVDD, ICL, and DTE at 1.0, showing that generalization is not limited to high-complexity regimes.

- **beth (high cardinality)**: On this large and high-cardinality dataset, CWAE (0.996) matches WAE and trails only slightly behind DTE (0.999), demonstrating scalability.

**Consistency distinguishes CWAE from other models.** CWAE ranks first or second on five of the eight datasets, spanning a spectrum of complexity levels. In contrast, models like RCA, SLAD, or DTE achieve top results only on one or two datasets and suffer from inconsistency elsewhere, as reflected by their higher average ranks.

These results affirm the core message of this work: context matters. In addition, we observed no systematic degradation or instability when evaluating samples from rare or unseen contexts, with performance approaching that of the non-contextual baseline as expected. When paired with effective context selection, CWAE

demonstrates scalable, generalizable, and state-of-the-art anomaly detection performance. The observed gains are consistent across a variety of domains and not limited to any single dataset characteristic, underscoring the broader value of integrating contextual information.

# 7 Ablation Studies

## 7.1 Benefits of Contextual Thresholding

Contextual anomaly detection models allow for **context-specific thresholds**, which offer a more nuanced decision boundary than a global threshold approach. This is particularly relevant in domains where the distribution of normal behavior varies significantly across groups defined by context features.

Figure 4 visualizes this idea by plotting the thresholds learned by CWAE on the *bank* dataset, conditioned on different context features. Each point corresponds to the threshold for a given group within a context feature. These thresholds $H_{\mathbf{C}_i}$ are obtained in the same manner described in Section 6.1.3.

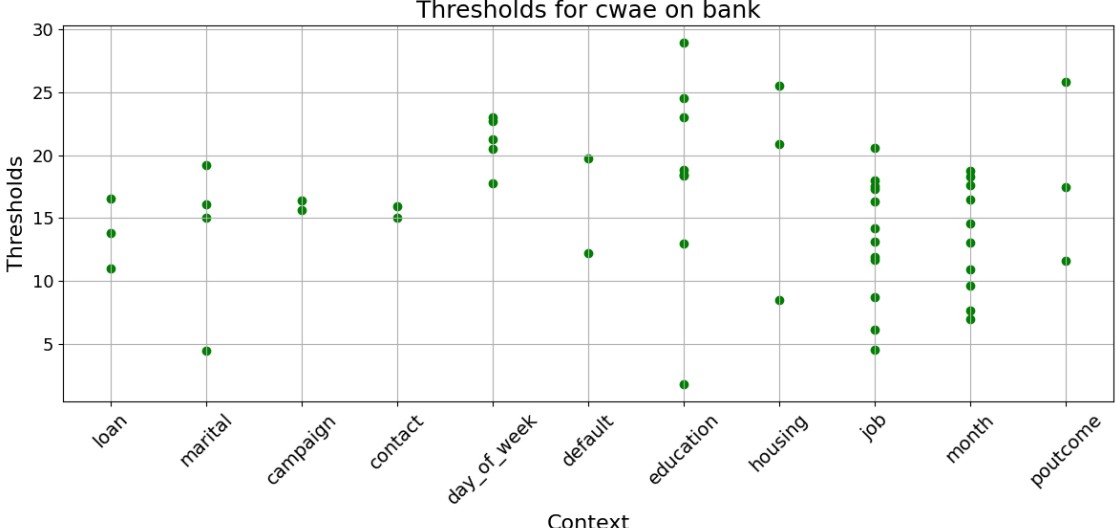

Figure 4: Thresholds for CWAE on the *bank* dataset, conditioned on different context features. Each point represents a threshold learned under a specific context group.

The primary takeaway from this figure is that **thresholds differ significantly across groups**, confirming our earlier claim that learning thresholds for each conditional distribution is desirable. This level of customization enables the model to detect context-dependent anomalies that may be missed under a single global threshold— particularly if such anomalies lie below the global cutoff but are relatively anomalous within their specific context.

It is important to **note that comparing raw threshold values across contexts or between contextual and non-contextual models is not strictly valid**. The threshold for WAE corresponds to $P(\mathbf{Y})$, while the thresholds for CWAE correspond to $P(\mathbf{Y} \mid \mathbf{C})$. Since the marginal $P(\mathbf{C})$ is not uniform across all datasets or features, and since scaling thresholds by $P(\mathbf{C})$ would alter the decision boundary, no fair normalization is possible without further assumptions.

Thus, the purpose of this visualization is not to support a direct comparison against WAE, but rather to illustrate that **contextual models naturally learn different thresholds for different conditions**. This leads to a more flexible and adaptive detection policy, consistent with the goals of contextual anomaly detection.

## 7.2 Upper Bound Performance with Optimal Context

In addition to evaluating CWAE under algorithmically selected context features, we assess its performance in the optimal context setting—where the context feature yielding the highest AUCROC score for each dataset is selected. This analysis reveals the potential upper bound of CWAE when context is ideally chosen. Table 5 presents the comparison where *CWAE Best* indicates the optimal context choices.

| Dataset | Best SOTA | | WAE | | CWAE | | CWAE Best | | CWAE Context | CWAE Best Context |
|---|---|---|---|---|---|---|---|---|---|---|
| | AUCROC | Rank | AUCROC | Rank | AUCROC | Rank | AUCROC | Rank | | |
| bank | 0.649 | 4 | 0.654 | 3 | 0.687 | 2 | **0.695** | **1** | loan | education |
| beth | **0.999** | **1** | 0.996 | 3 | 0.996 | 3 | 0.997 | 2 | no_ctx | argsNum |
| census | 0.701 | 3 | 0.676 | 4 | **0.743** | **1** | 0.743 | 1 | detailed_occupation_recode | detailed_occupation_recode |
| cmc | 0.760 | 2 | 0.702 | 4 | 0.751 | 3 | **0.787** | **1** | Husbands_education | Contraceptive_method_used |
| kdd | **0.939** | **1** | 0.674 | 3 | 0.639 | 4 | 0.872 | 2 | is_guest_login | service |
| lanl | **1.000** | **1** | 0.999 | 3 | 0.999 | 3 | **1.000** | **1** | SubjectLogonID | AuthenticationPackage |
| sf | 0.827 | 3 | 0.678 | 4 | 0.895 | 2 | **0.906** | **1** | C-class_flares_production_by_this_region | X-class_flares_production_by_this_region |
| spotify | 0.585 | 4 | 0.697 | 2 | 0.665 | 3 | **0.711** | **1** | loudness | valence |
| AVG | 0.808 | 2.375 | 0.760 | 3.25 | 0.797 | 2.625 | **0.839** | **1.25** | – | – |

Table 5: Comparison of AUCROC scores and ranks for WAE, CWAE, CWAE Best, and the best SOTA model per dataset. Bold and underlined values indicate the best AUCROC and best (lowest) rank.

**Optimal context significantly boosts CWAE performance.** On average, CWAE Best achieves an AUCROC of 0.839, outperforming the standard CWAE score of 0.797. It ranks first on 6 of the 8 datasets, compared to only 3 for the standard CWAE. This gap underscores the potential benefit of improved or learned context selection methods.

**CWAE Best consistently outperforms the non-contextual baseline.** Across all datasets, CWAE Best yields higher AUCROC scores than the non-contextual WAE baseline, often by substantial margins. For instance, on the *cmc* dataset, CWAE Best achieves 0.787 compared to WAE's 0.702, and on *kdd*, it reaches 0.872 versus WAE's 0.674. These improvements reinforce the core hypothesis of this work: that modeling context is a critical component for effective anomaly detection.

**CWAE Best surpasses all SOTA models on most datasets.** We aggregate the best scores from all individual baselines into a *Best SOTA* column to establish a performance reference. CWAE Best outperforms the Best SOTA model on 6 out of 8 datasets, including notable gains such as +0.046 on *bank*, +0.027 on *cmc*, and +0.079 on *sf*. On average, CWAE Best ranks first with a rank of 1.25, compared to Best SOTA's 2.375. Importantly, CWAE achieves this level of performance using a single unified architecture across all datasets, underscoring its value as a flexible, general-purpose anomaly detection model.

**CWAE Best performs well across the full range of dataset complexities.** Whether the dataset is low-complexity (e.g., lanl) or high-complexity (e.g., spotify), CWAE Best consistently delivers strong results. This suggests that with appropriate context, CWAE is broadly applicable across varied data regimes.

**Informative context selection drives performance gains.** The comparison between CWAE and CWAE Best isolates the benefit attributable specifically to selecting informative context features. Because CWAE Best enumerates all candidate context features for each dataset, it implicitly spans both informative and weak (or effectively uninformative) contexts. Empirically, poorly chosen contexts regress toward the non-contextual WAE baseline, while informative contexts yield substantial gains. As an illustrative consequence of the above, randomly selecting a context feature would, in expectation, fall within the same observed performance range, without introducing behavior beyond what is already captured by exhaustive enumeration. We therefore treat CWAE Best as an upper bound on context selection quality and the CWAE–CWAE Best gap as a direct measure of the value of improved context selection.

These results further support the importance of context selection. When given the optimal context, CWAE matches or surpasses state-of-the-art performance, demonstrating the potential of context-aware approaches. This suggests that future research should focus on improving context selection methods, including the development of learnable strategies that can consistently achieve performance close to this upper bound in practice.

### 7.3 Quantifying the Value of Context

A central question in this research is whether incorporating contextual information improves anomaly detection performance not only across datasets of varying complexity, but also different model architectures. To investigate, we ran additional experiments with other conditional models from the AD literature. The selected models are listed below:

- **Conditional Wasserstein Autoencoder (CWAE)**, described in Section 5.1.

- **Conditional Variational Autoencoder (CVAE)** Pol et al. (2020), which models the conditional distribution $P_\theta(\mathbf{Y} \mid \mathbf{C})$ of observations given contextual variables using a deep generative model with learned feature-wise reconstruction variance.

- **Conditional Mixture Density Network (CMDN)** Dai et al. (2025), which learns a neural network-parameterized Gaussian mixture model to capture $P(\mathbf{Y} \mid \mathbf{C})$, enabling mixture parameters to adapt dynamically based on context.

- **Contextual Anomaly Detection using Isolation Forest (CADI)** Yepmo et al. (2024), which extends the isolation forest algorithm with density-aware splits to jointly detect anomalies and explain them relative to local contextual clusters.

All models are implemented with only minor modifications from their original formulations, using the same embedding layers outlined in Section 5.1 (see Appendix A for more details). For each model-dataset pair, we systematically iterated over all candidate features as potential context, designating exactly one feature as context in each run and treating the remainder as content. Performance was compared to a *no-context baseline* in which all features are treated as content. This controlled setting enables a direct, interpretable measure of how both context and model architecture contribute to performance improvements.

To quantify the benefits of contextual learning, we report the average improvement in AUCROC across datasets for each model, comparing the best-performing context configuration to its non-contextual counterpart. All contextual models outperform their non-contextual baselines on average, often by a substantial margin. The CWAE model achieves the highest average improvement of +11.69%, followed by CMDN (+7.47%), CVAE (+6.21%), and CADI (+4.18%). These consistent gains support the hypothesis that leveraging context enhances a model's ability to detect anomalies across architectures, especially when the contextual signal is informative.

Given its strong empirical performance, we selected CWAE as the primary model for further analysis and methodological development. We treat CWAE as a representative upper bound for context-aware models in our framework, using it as a proxy to explore context selection strategies and to benchmark the overall value of contextual information in anomaly detection.

## 8 Conclusion

This work establishes *contextual learning* as a general paradigm for anomaly detection in tabular data. The proposed framework provides a unified probabilistic formulation, a principled context selection strategy, and a theoretical foundation for modeling conditional distributions $P(\mathbf{Y} \mid \mathbf{C})$. Through variance decomposition and discriminative learning principles, we show that conditioning effectively isolates intra-context variability while mitigating noise from inter-context differences, leading to more robust and precise anomaly detection.

We instantiate this paradigm through CWAE, a lightweight generative model that operationalizes the contextual learning framework for tabular domains. Across diverse datasets spanning finance, cybersecurity, demographics, and network intrusion detection, the CWAE instantiation consistently outperforms both its non-contextual baseline (WAE) and SOTA unconditional models, demonstrating the practical effectiveness and generality of contextual learning.

Future research will focus on extending contextual learning to multi-context conditioning, developing learnable context discovery mechanisms, and applying the framework to other data modalities such as text, time series, and multimodal environments.

## 9   Broader Impact and Limitations

This work proposes a contextual learning framework for unsupervised anomaly detection in tabular data, with primary intended applications in areas such as cybersecurity, fraud detection, and system monitoring. By modeling conditional distributions rather than a single global data distribution, the approach has the potential to improve detection accuracy in heterogeneous environments. At the same time, the use of context-aware anomaly detection raises several broader impact considerations that merit discussion.

**Potential for Bias and Context Selection.**   A central component of the proposed framework is the selection of context features that condition the learned notion of normal behavior. While contextual modeling can reduce certain forms of bias by avoiding a single global decision boundary, it also introduces the risk that chosen context features may correlate with sensitive attributes such as race, gender, age, or socioeconomic status. In such cases, the model may learn different anomaly score distributions or thresholds across demographic groups, potentially resulting in disparate false positive or false negative rates. This risk is not unique to the proposed method but is particularly salient because context explicitly influences decision boundaries. Practitioners deploying contextual anomaly detection systems should therefore carefully audit candidate context features, assess correlations with sensitive attributes, and evaluate group-wise performance to identify unintended discriminatory effects.

**Dual-Use Considerations.**   Like most advances in anomaly detection, the proposed framework has dual-use potential. Improved anomaly detection can yield clear societal benefits when applied to domains such as fraud prevention, network security, and system reliability. However, the same techniques could also be applied in ways that raise ethical concerns, for example in large-scale surveillance, behavioral monitoring, or profiling without appropriate safeguards. This work does not prescribe specific deployment contexts, and responsible use ultimately depends on organizational policies, legal frameworks, and oversight mechanisms governing how anomaly detection systems are applied.

**Data Use, Transparency, and Reproducibility.**   From an ethical research practice perspective, this work relies exclusively on publicly available benchmark datasets that have been widely used in prior anomaly detection research, and no proprietary or personally identifiable data are introduced. Implementation details, experimental settings, and additional analyses are provided in the appendix to support reproducibility and facilitate independent verification. While reproducibility does not by itself guarantee ethical deployment, transparency in data usage and evaluation is an important prerequisite for responsible research and for enabling follow-up work that examines robustness, bias, and broader impacts more closely.

**Behavior Under Rare Contexts.**   For rare context values with limited support (e.g., fewer than 10 samples), the model may not reliably learn context-specific structure. In these cases, contextual thresholds are estimated conservatively through the normalization procedure described in Section 6.1.3. Because anomaly scores are normalized relative to per-context training maxima, rare contexts do not induce arbitrarily tight decision boundaries. As a result, the detector relies more heavily on content-level structure rather than memorizing small context-specific patterns, reducing the risk of overfitting in low-support regimes but also limiting the benefits of contextual conditioning when data are sparse.

**Sensitivity to Validation Data Quality.**   The proposed context selection procedure relies on a small validation subset to rank candidate context features based on early-stage generalization performance. This approach assumes that the validation data is predominantly representative of normal operating conditions, which is standard in unsupervised anomaly detection but may not hold in all deployment scenarios. When the validation set is heavily contaminated with anomalies, validation-based context ranking may become less reliable, potentially leading to uninformative or even detrimental context selection. Addressing such settings may require contamination-aware or weakly supervised extensions that explicitly account for noise or anomalous samples in the validation data. Developing context selection strategies that remain robust under substantial validation-set contamination is an important direction for future work.

Overall, this work is intended as a methodological contribution to anomaly detection research. The considerations outlined above highlight the importance of careful context selection, auditing, and responsible

deployment when applying contextual anomaly detection methods in real-world settings, particularly those involving human-centered or high-stakes decisions.

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

# A  Appendix

This appendix provides comprehensive implementation and evaluation details to help facilitate the reproducibility of our experimental results.

## A.1  Model Architectures

**CWAE Architecture Details**  The Conditional Wasserstein Autoencoder (CWAE) implements the following architecture:

**Embedding Layers:**

- **Type**: Trainable embedding layers initialized randomly
- **Dimension**: 16-dimensional embeddings for categorical features
- **Vocabulary Size**: Determined by unique values in training data for each feature, plus 1 for handling unknown values at test time
- **Unknown Value Handling**: Reserved index initialized to maximum values across embedding dimensions

**Encoder:**

- **Input Layer**: Context embeddings ($\mathbb{R}^c$) concatenated with content embeddings ($\mathbb{R}^{d-c}$)
- **Hidden Layer 1**: Linear transformation from $(c + d - c)$ to 128 dimensions with ReLU activation
- **Hidden Layer 2**: Linear transformation from 128 to 64 dimensions with ReLU activation
- **Latent Layer**: Linear transformation from 64 to $z_{dim}$ dimensions, where $z_{dim} = 64$ by default
- **Regularization**: No bias terms used in linear layers

**Decoder:**  For each content feature $i \in \{1, \ldots, d - c\}$:

- **Input**: Latent representation $z$ concatenated with context embeddings, dimension $\mathbb{R}^{z_{dim}+c}$
- **Hidden Layer 1**: Linear transformation from $(z_{dim} + c)$ to 64 dimensions with ReLU activation
- **Hidden Layer 2**: Linear transformation from 64 to 128 dimensions with ReLU activation
- **Output Layer**: Linear transformation from 128 to vocabulary size for feature $i$, producing logits for cross-entropy loss computation
- **Regularization**: No bias terms used in linear layers

**CVAE Architecture Details**  The Conditional Variational Autoencoder extends CWAE with probabilistic latent variables:

- **Encoder**: Same architecture as CWAE encoder, but outputs both mean $\mu$ and log-variance $\log \sigma^2$ for the latent distribution
- **Latent Sampling**: Reparameterization trick used: $z = \mu + \sigma \odot \epsilon$, where $\epsilon \sim \mathcal{N}(0, I)$
- **Decoder**: Identical architecture to CWAE decoder
- **KL Divergence**: Computed as $D_{KL}(q(z|x,c)||p(z)) = -\frac{1}{2} \sum_{j=1}^{z_{dim}} (1 + \log \sigma_j^2 - \mu_j^2 - \sigma_j^2)$

**CMDN Architecture Details**  The Conditional Mixture Density Network models output distributions as Gaussian mixtures:

- **Encoder**: Similar architecture to CWAE with context conditioning

- **Output**: Predicts mixture parameters $(\pi_k, \mu_k, \sigma_k)$ for $K = 5$ mixture components

- **Mixture Weights**: Normalized using softmax: $\pi_k = \frac{\exp(\alpha_k)}{\sum_{j=1}^{K} \exp(\alpha_j)}$

- **Loss Function**: Negative log-likelihood of the Gaussian mixture: $-\log \sum_{k=1}^{K} \pi_k \mathcal{N}(x|\mu_k, \sigma_k^2)$

## A.2 Training Hyperparameters

**Default Training Configuration**  All models use consistent hyperparameters across experiments. This is shown in Table 6.

| Parameter | Value |
|---|---|
| Learning Rate | 0.001 |
| Optimizer | Adam ($\beta_1 = 0.9$, $\beta_2 = 0.999$, $\epsilon = 10^{-8}$) |
| Batch Size | 2048 |
| Maximum Epochs | 25 |
| Early Stopping Threshold | 0.0001 (on training loss) |
| Weight Decay | 0 (no L2 regularization) |
| Gradient Clipping | None |
| Random Seed | 43 |

Table 6: Default training hyperparameters used across all experiments.

**Model-Specific Adjustments**

- **CWAE**: MMD regularization weight $\lambda = 1.0$

- **CVAE**: KL divergence weight $\beta = 1.0$ (standard VAE formulation)

- **CMDN**: Early stopping threshold = -20.0 (adapted to negative log-likelihood scale)

- **CADI**: Uses 100 isolation trees with automatic contamination parameter selection

## A.3 Computational Environment

**Hardware Specifications**  Experiments were conducted on the following hardware:

- **GPU**: NVIDIA A100 GPU (40GB memory) with CUDA 12.4 support

- **CPU**: Multi-core processor for parallel data loading (4 workers)

- **RAM**: Sufficient for batch size of 2048 across all datasets

- **Storage**: SSD storage for efficient data I/O operations

**Software Dependencies**  Experiments were conducted with the following software dependencies. These are shown in Table 7

| Package | Version |
|---|---|
| Python | 3.8+ |
| PyTorch | 2.6.0+cu124 |
| CUDA | 12.4 |
| cuDNN | 9.1.0.70 |
| NumPy | 2.0.2 |
| Pandas | 2.3.0 |
| Scikit-learn | 1.6.1 |
| Matplotlib | 3.9.4 |
| Seaborn | 0.13.2 |

Table 7: Software dependencies and versions used in experiments.

### A.4   Joint Validation Loss Computation

**Motivation**   To enable fair comparison between joint models (learning $p(x, y)$) and conditional models (learning $p(x|y)$), we convert all models to evaluate on the joint probability $p(x, y)$, as these distributions have different scales.

**Conversion Methodology**   For contextual models learning $p(x|y)$, we apply the probability chain rule:

$$p(x, y) = p(x|y) \cdot p(y) \tag{2}$$

In log-space, this becomes:

$$-\log p(x, y) = -\log p(x|y) - \log p(y) \tag{3}$$

**Marginal Probability Estimation:**

1. Estimate $p(y)$ empirically from training data: $\hat{p}(y = c) = \frac{\text{count}(c)}{N}$

2. Compute log probabilities: $\log \hat{p}(y = c) = \log(\text{count}(c)) - \log(N)$

3. Handle unseen values: Use minimum observed probability as a fallback for rare categories

**Sample-wise Correction:**   For each validation sample $(x_i, y_i)$:

$$\text{joint\_loss}_i = \text{model\_loss}_i + \left( -\sum_{j \in \text{context}} \log \hat{p}(y_{ij}) \right) \tag{4}$$

**Batch-wise Average:**

$$\text{Joint Validation Loss} = \frac{1}{B} \sum_{i=1}^{B} \text{joint\_loss}_i \tag{5}$$

where $B$ is the number of validation batches (limited to 10 batches for computational efficiency).

### A.5   Datasets

### A.5.1   Data Format

All datasets were preprocessed into a standardized format:

- **Anomaly Label**: Binary column with values 0 (normal) and 1 (anomaly)

- **Feature Encoding**: All features encoded as categorical integer indices

- **Missing Values**: Handled via imputation or exclusion prior to experiments

### A.5.2    Feature Encoding

1. **Categorical Encoding**: All features mapped to integer indices using label encoding

2. **Index 0 Reserved**: Designated for unknown or unseen categorical values during inference

3. **Vocabulary Construction**: Built exclusively from training data; validation and test sets may contain previously unseen values

4. **No Numerical Scaling**: Not required as features are processed through trainable embedding layers

### A.5.3    Context/Content Feature Assignment

For each dataset and context configuration:

1. **Context Features**: Selected feature(s) used for conditional modeling

2. **Content Features**: Remaining features to be reconstructed by the model

3. **Column Organization**: Features organized as context features followed by content features

4. **No-Context Baseline**: All features treated as content (empty context set)

### A.5.4    Train/Validation/Test Splits

All experiments use fixed pre-determined data splits:

- **Training Set**: Contains only normal samples (anomaly=0)

- **Validation Set**: Contains only normal samples, used for context selection

- **Test Set**: Contains both normal and anomalous samples for evaluation

- **Split Ratios**: Approximately 60% train, 20% validation, 20% test

- **Split Strategy**: Random but fixed across all experiments to ensure fair comparison

### A.5.5    Reproducibility Measures

- **Fixed Random Seed**: Seed value of 43 used for all random operations

- **Data Loading**: Training data shuffled; validation and test data processed in fixed order

- **Consistent Splits**: Same train/validation/test partitions used across all model comparisons

### A.6    Evaluation Metrics

### A.6.1    Anomaly Scoring

For reconstruction-based models (CWAE, CVAE, CMDN), the anomaly score is computed as:

$$\text{score}(x, c) = \sum_{i=1}^{d-k} \text{CE}(y_i, \hat{y}_i) \tag{6}$$

where CE denotes cross-entropy loss, $y_i$ represents the true content feature value, $\hat{y}_i$ is the model's reconstruction, and $d - k$ is the number of content features.

### A.6.2 Context-Specific Thresholding

1. **Training Threshold**: For each context value $c$, compute maximum training score: $H_c = \max_{(x,c)\in\text{train}} \text{score}(x, c)$

2. **Normalization**: Normalize test scores relative to training threshold: $R_c(x) = \frac{\text{score}(x,c)}{H_c}$

3. **Global Threshold Search**: Grid search over 100 uniformly-spaced thresholds in range $[0, \max(R_c)]$

4. **AUC-ROC Computation**: Use normalized ratios as continuous anomaly scores

### A.6.3 Performance Metrics

- **AUC-ROC**: Area under receiver operating characteristic curve

- **AUC-PR**: Area under precision-recall curve

- **F1-Score**: Harmonic mean of precision and recall at optimal threshold

- **Precision**: $\frac{TP}{TP+FP}$ at optimal threshold

- **Recall**: $\frac{TP}{TP+FN}$ at optimal threshold

Optimal threshold determined by maximizing F1-score on the validation set.

## A.7 Context Selection Procedure

### A.7.1 Bilevel Optimization Framework

**Outer Loop (Context Selection):**

- **Candidate Set**: All individual features tested as single-feature contexts

- **Baseline**: No-context configuration included for comparison

- **Evaluation Protocol**: Train each candidate for 1 epoch and compute joint validation loss

**Inner Loop (Model Training):**

- **Training Duration**: Single epoch for context selection (early-epoch proxy)

- **Objective**: Minimize training loss using standard hyperparameters

- **No Validation-Based Training**: Validation set only used for evaluation, not for early stopping during context selection

**Selection Criterion:**

$$c^* = \arg\min_{c\in\mathcal{C}} \mathbb{E}_{(x,y)\sim\mathcal{D}_{\text{val}}}[-\log p(x, y|\theta_c)] \tag{7}$$

where $\theta_c$ represents model parameters trained with context feature $c$, and $\mathcal{C}$ is the set of candidate context features.

### A.7.2 Early Epoch Justification

Single-epoch evaluation is motivated by:

- **Computational Efficiency**: Reduces context selection time by approximately 96% (25× speedup compared to full training)

- **Relative Ranking**: Early training dynamics strongly correlate with final performance rankings (Liu et al., 2019)

- **Empirical Validation**: Figure 3 demonstrates that the early-epoch proxy achieves better final performance on 5 of 8 datasets

- **Established Practice**: Similar approaches used successfully in neural architecture search (Bender et al., 2018), model pruning (You et al., 2020), and noisy label learning (Liu et al., 2020)

### A.8 Runtime Analysis

### A.8.1 Training Time

Approximate training times per model/dataset combination (25 epochs on NVIDIA A100 GPU) shown in Table 8.

| Dataset | Training Time | Samples | Time/Epoch |
|---------|--------------:|--------:|-----------:|
| bank    | 5 min   | 41,188    | 12 sec  |
| beth    | 45 min  | 1,141,078 | 108 sec |
| census  | 12 min  | 299,285   | 29 sec  |
| cmc     | 2 min   | 1,473     | 5 sec   |
| kdd     | 40 min  | 1,014,535 | 96 sec  |
| lanl    | 1.5 hrs | 2,542,727 | 216 sec |
| sf      | 2 min   | 1,066     | 5 sec   |
| spotify | 8 min   | 113,550   | 19 sec  |

Table 8: Approximate training times for CWAE on each dataset.

### A.8.2 Inference Time

Per-sample anomaly score computation:

- **Forward Pass**: Less than 1 millisecond per sample with batch size 2048

- **GPU Acceleration**: Significantly faster than CPU-based inference

- **Deterministic Scoring**: CWAE requires single forward pass (no sampling) unlike stochastic models (CVAE)

### A.8.3 Context Selection Time

For each dataset with $N$ candidate context features:

- **Single Context Training**: 1 epoch ≈ 4% of full training time

- **Total Context Selection**: $N \times$ (1-epoch time) + validation evaluation time

- **Example (census)**: 38 features × 29 sec/epoch ≈ 18.4 minutes

- **Validation Evaluation**: Less than 1 minute per context (evaluated on 10 batches)

