# OpenReview forum: "Contextual Learning for Anomaly Detection in Tabular Data"
_TMLR — Accepted by TMLR_

### Review · Reviewer_yuXt · 2025-11-26

**Summary Of Contributions:**

This paper presents a contextual learning framework for anomaly detection in tabular data, motivated by the observation that global joint-distribution modeling often fails to capture heterogeneous behavioral patterns. Rather than modeling P(X), the authors focus on P(Y|C) by separating context variables from content features and learning conditional distributions. They introduce (i) a probabilistic formulation based on variance decomposition and discriminative-learning principles, (ii) a bilevel context-selection strategy via single-epoch validation loss, and (iii) a Contextual Wasserstein Autoencoder model. Experiments across eight datasets demonstrate that contextual learning improves over global models and that CWAE outperforms non-contextual WAE and other SOTA anomaly-detection baselines on average.

Strengths:
- CWAE shows consistent gains over WAE and often surpasses strong non-contextual SOTA baselines, supported by AUCROC tables and average-rank comparisons.
- The theoretical framing is well-motivated, clearly linking conditional density estimation with variance reduction, offering conceptual insight into why contextual modeling helps.
- The bilevel context-selection strategy is simple, computationally efficient, and empirically validated through early-epoch behavior.

Weaknesses:
- The role of context vs. content is under-formalized. While the authors claim generalizability to multi-context conditioning, implications such as feature interactions, combinatorial search cost, and risk of degradation are not thoroughly examined.
- The thresholding and contextual score normalization strategy is engineered but not statistically grounded—there is little discussion of false-alarm rate, stability across context-sizes, or comparability with baselines using different score distributions.
- Some implementation decisions appear insufficiently justified—for instance why one epoch reliably predicts context quality, why embeddings are concatenated before encoding, and how embedding representations are initialized or optimized.

**Additional Comments:**

- Could CWAE generalize to other autoencoder families (e.g., convolutional, diffusion, transformer-style) and to non-tabular domains such as images or logs?
- How would the framework behave under multi-context conditioning, continuous context variables, or learned context embeddings?

**Audience:**

Yes

**Audience Explanation:**

Tabular unsupervised anomaly detection is an active area, and contextual modeling is applicable across many fields. The framework is general, interpretable, model-agnostic, and applicable to existing architectures.

**Broader Impact Concerns:**

No concern.

**Claims And Evidence:**

Yes

**Claims Explanation:**

Performance improvements are documented across eight datasets with AUCROC, ranked comparison tables, and multiple contextual baselines. The theoretical claims are reasonably supported.

**Requested Changes:**

- Elaborate on how context/content features are partitioned, and discuss feasibility of multi-context conditioning.
- Strengthen justification for the complexity ranking metrics and provide more literature linkage or comparative baselines.
- Improve motivation for choosing the Wasserstein approach over VAE/CVAE and other reconstruction-based models.
- Reference related literature from statistical process control and multivariate anomaly detection (e.g., control charts, Hotelling-T²/Mahalanobis distance, autoencoder-based MSPC) to position contributions more clearly within the broader methodological landscape.
- Provide further ablation on one-epoch context selection to verify robustness of early-training ranking signals.

---

> ### Author Response · Authors · 2026-01-07
> **Author response to Reviewer yuXt 1/2**
>
> Many thanks for the review and the helpful suggestions. We will respond to each requested change and all comments individually below.
>
> ---
>
> ## Requested Change 1
>
> >Elaborate on how context/content features are partitioned, and discuss feasibility of multi-context conditioning.
>
> We will clarify earlier in the paper how context and content features are partitioned in the proposed framework. Specifically, we define context features as conditioning variables that index heterogeneous normal behavior, while content features are the remaining variables whose conditional distribution is modeled. In the tabular setting considered here, candidate context features are drawn from the observed feature set and selected automatically via validation loss, with all remaining features treated as content. This partition is dataset-dependent and does not assume prior semantic knowledge; selecting an uninformative context naturally recovers a global baseline.
>
> Regarding multi-context conditioning, while the framework can in principle condition on multiple context variables, its practical feasibility depends on data regime and context structure. Conditioning on multiple features induces a combinatorial growth in context cells, which can lead to severe data sparsity and unstable threshold estimation in unsupervised settings. Moreover, validation-based context selection becomes increasingly noisy as context dimensionality grows, particularly when normal data are limited. For these reasons, we restrict our empirical study to single-feature contexts, where context-specific behavior can be estimated reliably and interpreted clearly. We will make these practical considerations explicit in the revisions.
>
> ---
>
> ## Requested Change 2
>
> >Strengthen justification for the complexity ranking metrics and provide more literature linkage or comparative baselines.
>
> We will strengthen the motivation for the dataset complexity metrics by clarifying their role as diagnostic descriptors rather than evaluation criteria. The intent of these metrics is not to claim a new complexity measure, but to contextualize results across datasets that differ substantially in structure, heterogeneity, and anomaly separability.
>
> We will more explicitly link these metrics to prior anomaly detection literature, particularly Pang et al. ([2021a, Homophily outlier detection in non-IID categorical data](https://arxiv.org/pdf/2103.11516)), and clarify how each metric captures a known difficulty factor (e.g., homophily, categorical skew, inseparability, noise). We will also clarify why these metrics are complementary to basic dataset statistics such as size, dimensionality, or cardinality, which often fail to reflect anomaly detection difficulty in tabular settings.
>
> Importantly, we will emphasize that no claims in the paper depend on these metrics, and that all conclusions are drawn directly from empirical performance comparisons. The complexity rankings are provided to aid interpretation rather than to support any methodological claim.
>
> ---
>
> ## Requested Change 3
>
> >Improve motivation for choosing the Wasserstein approach over VAE/CVAE and other reconstruction-based models.
>
> We will expand Section 5.4 to emphasize that deterministic encoding and stable inference are particularly aligned with the goals of contextual anomaly detection, where consistent scoring and low variance are critical, especially when anomaly scores must be compared across heterogeneous contexts. This design choice is complementary to the broader contextual learning framework and not essential to its validity.
>
> ---
>
> ## Requested Change 4
>
> > Reference related literature from statistical process control and multivariate anomaly detection (e.g., control charts, Hotelling-T²/Mahalanobis distance, autoencoder-based MSPC) to position contributions more clearly within the broader methodological landscape.
>
> We agree that this connection strengthens positioning and helps situate the paper within a broader methodological lineage while clarifying its novelty and scope. We will expand the related work section to explicitly reference classical statistical process control (SPC) and multivariate monitoring methods, including Hotelling’s T², Mahalanobis-distance-based detectors, and autoencoder-based MSPC approaches.
>
> We will highlight the conceptual parallel between SPC-style conditional baselines (e.g., operating regimes, control states) and our notion of context-conditioned modeling, while clarifying that classical methods typically assume Gaussianity, linear structure, or known regimes. Our framework generalizes this intuition to high-dimensional, categorical, and heterogeneous tabular data without requiring explicit regime labels.

---

> ### Author Response · Authors · 2026-01-07
> **Author response to Reviewer yuXt 2/2**
>
> ## Requested Change 5
>
> >Provide further ablation on one-epoch context selection to verify robustness of early-training ranking signals.
>
> We acknowledge this concern and will make explicit that the goal of the one-epoch proxy is relative ranking of candidate contexts rather than accurate performance estimation. We will strengthen the discussion by drawing clearer parallels to prior work showing that early training dynamics reliably preserve model rankings in architecture search, pruning, and data valuation.
>
> We will also clarify that Figure 3 already provides empirical evidence that the early-epoch proxy is sufficient for effective selection on the majority of datasets, and that in cases where context is uninformative, the method defaults to the no-context baseline.
>
> ---
>
> ## Additional Comment 1
>
> >Could CWAE generalize to other autoencoder families (e.g., convolutional, diffusion, transformer-style) and to non-tabular domains such as images or logs?
>
> We thank the reviewer for this question. The proposed contextual learning framework is not inherently tied to CWAE or to tabular data; CWAE serves as a simple and effective instantiation that allows us to isolate the effects of context conditioning in a controlled setting. More generally, the framework only assumes a conditional encoder–decoder structure with context-conditioned scoring, and in principle can be instantiated with other autoencoder families, including convolutional, transformer-based, or diffusion-style architectures.
>
> Similarly, while this work focuses on tabular data, the formulation itself does not preclude application to other modalities such as images or logs, provided that meaningful context variables and appropriate encoders are available. Exploring such extensions would require additional architectural and experimental considerations and is therefore beyond the scope of the present study. We view generalization across model families and modalities as a promising direction for future work.
>
> ---
>
> ## Additional Comment 2
>
> >How would the framework behave under multi-context conditioning, continuous context variables, or learned context embeddings?
>
> We appreciate this thoughtful question. At a conceptual level, the framework can accommodate multi-context conditioning, continuous context variables, or learned context representations. However, the practical behavior of such extensions depends strongly on data regime and selection stability. Conditioning on multiple or continuous contexts can substantially increase the effective context space, leading to data sparsity and unreliable estimation in unsupervised settings.
>
> For this reason, we focus in this work on single-feature, discrete contexts with simple learned embeddings, which allows us to isolate the benefit of contextual conditioning while maintaining interpretability and stability. Richer context representations, including multi-context conditioning or learned context embeddings, introduce additional modeling choices and trade-offs that are orthogonal to the core contribution of this paper. We therefore leave a systematic exploration of these extensions to future work.

---

### Review · Reviewer_nSmt · 2025-12-24

**Summary Of Contributions:**

The paper proposes a contextual learning framework for anomaly detection in tabular data that models conditional distributions P(Y∣C) instead of a single global joint distribution P(X), instantiates it via a Conditional Wasserstein Autoencoder (CWAE), and shows that contextual models can surpass strong non-contextual baselines on eight benchmarks. The core contributions are:

i) Formalizing anomaly detection with a split between context features C and content features Y, arguing for modeling P(Y∣C) with context-specific thresholds instead of a global P(X)-based detector.
​
ii) Using a variance decomposition (law of total variance) and generative-vs-discriminative arguments (misspecification, curse of dimensionality) to justify why conditioning on context can reduce variance and improve anomaly detection.
​
iii) Proposing a bilevel optimization approach that selects a single context feature by minimizing early joint validation loss (after one epoch) across candidates, using only (mostly normal) validation data and no anomaly labels.
​
iv) Introducing CWAE, a conditional WAE with embeddings for tabular categorical features, deterministic encoder-decoder, MMD regularization, and per-context thresholding for anomaly scoring.
​
v) Empirical evaluation is done on benchmarks CWAE against seven non-contextual deep AD baselines over eight tabular datasets, reporting that CWAE attains the best average AUCROC (0.797) and lowest average rank (2.75), improving over its non-contextual WAE counterpart (0.760, rank 3.875).
​
vi) Providing ablations on contextual thresholding, optimal context choice (CWAE Best reaching average AUCROC 0.839 and rank 1.25, outperforming best SOTA on 6/8 datasets), and additional conditional baselines (CVAE, CMDN, CADI) to quantify the value of context.


Key Strengths:

i) The paper addresses a significant, practical challenge: heterogeneity in real-world tabular data.

ii) The framework is well-motivated theoretically (variance reduction, discriminative advantages) and practically (scalable context selection).

iii) The experimental design is rigorous, using diverse datasets and comparing against multiple strong baselines.

iv) The proposed CWAE is simple and effective, serving as a clear proof-of-concept for the broader framework.


Key Weaknesses:

i) The theoretical connection between variance reduction and improved anomaly detection, while intuitive, could be elaborated more formally.

ii) The ablation studies, while present, could more deeply analyze the individual contribution of the bilevel optimization strategy versus the conditional modeling itself.

iii) Context selection is restricted to single-feature conditioning, justified mostly for simplicity and interpretability; multi-feature contexts are only discussed as future work despite being central to many realistic settings.

iv) The proposed context selection relies on a mostly clean validation set and an early-epoch proxy; robustness to contamination, stability across seeds, and sensitivity to the 1 epoch choice are not fully stress-tested.

v) The evaluation introduces a context-normalized scoring/thresholding scheme (max training loss per context), which may inflate robustness in some cases and is potentially sensitive to extremes; it deserves deeper justification and alternatives.

vi) The framework is mostly evaluated with single-feature context and primarily categorical-encoding pipelines; generality to continuous/tabular mixed-type settings is not demonstrated convincingly.

vii) Baseline comparisons may be uneven: many baselines are used with default settings and without contextual variants, so the SOTA claim is strongest for the CWAE-vs-unconditional setting, but less clear for contextualized competitors.
​
viii) Evaluation focuses almost entirely on AUCROC, which can be misleading under heavy class imbalance; AUC-PR and operational metrics are mentioned in the appendix but are not central to the main narrative.
​
ix) The framework's dependence on the availability of meaningful context candidates is not stress-tested in adversarial settings (e.g., when candidate context features are weak, redundant, or noisy).

x) It would have been better if they had shared the repository, respecting the journal guidelines.

**Audience:**

Yes

**Audience Explanation:**

The audience of TMLR  would be interested in knowing the findings of this paper. The paper touches on

i) Relevance to Core ML: The paper tackles a fundamental issue in density estimation and generative modeling, how to handle heterogeneous data by advocating for a shift from joint to conditional modeling. The theoretical discussion connects to classic generative vs. discriminative trade-offs.

ii) Practical Impact: Anomaly detection is a critical task in high-stakes domains like cybersecurity and finance, where the paper's data originates. The proposed method offers a principled way to improve detection in real-world, non-IID tabular data, which is a pervasive and challenging data type.

iii) Novelty: The integration of a bilevel optimization for unsupervised context feature selection is a novel and practical contribution. The CWAE model, while simple, effectively demonstrates the framework's viability.

Therefore, the findings would be of significant interest to those working on anomaly detection, representation learning, and the application of ML to structured data.

**Broader Impact Concerns:**

The paper presents a methodological advancement in anomaly detection with primarily positive potential impacts (improved security, fraud detection). However, a Broader Impact Statement should be added to address the following:

i) Potential for Bias: The method's performance hinges on context selection. If the chosen context feature correlates with sensitive attributes (e.g., race, gender), the model could learn to flag anomalies differently for different demographic groups, potentially leading to discriminatory outcomes. The paper should briefly discuss this risk and the importance of auditing context features.

ii) Dual-Use Potential: Like all anomaly detection research, more effective tools could be used for both beneficial (network security) and harmful (surveillance) purposes. Acknowledging this duality is standard.

iii) Data and Reproducibility: The statement could highlight the use of publicly available benchmark datasets and the provided appendix as positive steps for reproducibility and ethical research practice. Currently, the absence of such a statement is a minor oversight.

**Claims And Evidence:**

Yes

**Claims Explanation:**

The claims are strongly supported by a combination of theoretical reasoning and comprehensive empirical evidence.

i) Theoretical Claims: The arguments about variance reduction (via the law of total variance) and the benefits of discriminative/conditional modeling are well-established in statistical learning literature and are correctly applied to the anomaly detection problem.

ii) Empirical Claims: The experimental section is thorough. The use of eight diverse datasets covers various domains and complexities. The comparison includes a relevant non-contextual baseline (WAE) and several SOTA models. Results (AUCROC, ranks) are clearly presented in tables and show consistent gains for the contextual approach. Ablation studies on contextual thresholding and optimal context upper bounds provide additional convincing evidence.

iii) Clarity: The methodology for context selection, model training, and evaluation (including the nuanced contextual AUCROC calculation) is described in sufficient detail for reproducibility. Figures effectively illustrate key points like threshold variation and validation loss curves.

**Requested Changes:**

i) Strengthen the theoretical section (Section 4). The link between reduced conditional variance Var(Y∣C) and improved anomaly detection performance is argued intuitively but not formally proven. A more formal discussion or a simple theorem/bound connecting variance to anomaly score separability would significantly bolster the paper's foundation.

ii) While Algorithm 1 is evaluated, a more direct ablation is needed to disentangle the gain from using any reasonable context vs. the gain from the optimally selected context via the proposed method. Consider adding a small experiment comparing performance using a randomly chosen context feature versus the one selected by the bilevel method across multiple runs/datasets.

iii) In Section 5.3 and Algorithm 1, clarify that the validation set is assumed to contain only normal samples, which is a standard but important assumption for unsupervised AD. This is mentioned later but should be explicit in the algorithm description.

---

> ### Author Response · Authors · 2026-01-07
> **Author response to Reviewer nSmt 1/2**
>
> Many thanks for the review and the helpful suggestions. We will respond to each requested change individually below.
>
> ---
>
> ## Requested Change 1
>
> >Strengthen the theoretical section (Section 4). The link between reduced conditional variance Var(Y∣C) and improved anomaly detection performance is argued intuitively but not formally proven. A more formal discussion or a simple theorem/bound connecting variance to anomaly score separability would significantly bolster the paper's foundation.
>
> We appreciate this suggestion and agree that the theoretical connection can be made more explicit. We will strengthen Section 4 by adding a concise, proposition-style argument that formalizes the relationship between reduced conditional variance and anomaly-score separability. In particular, under standard concentration assumptions on reconstruction error, we will show that conditioning on context yields tighter error distributions for normal samples, which in turn increases the separation between normal and anomalous score distributions. This addition is intended to complement the existing variance-decomposition intuition with a minimal but formal discriminative argument.
>
> ---
>
> ## Requested Change 2
>
> >While Algorithm 1 is evaluated, a more direct ablation is needed to disentangle the gain from using any reasonable context vs. the gain from the optimally selected context via the proposed method. Consider adding a small experiment comparing performance using a randomly chosen context feature versus the one selected by the bilevel method across multiple runs/datasets.
>
> We agree that it is important to disentangle the benefit of conditioning per se from the benefit of selecting an informative context. We will revise the paper to make this decomposition explicit, particularly in Sections 6.2 and 7.2.
>
> First, the benefit of conditioning itself is isolated by comparing CWAE to its non-contextual counterpart WAE. These models share the same architecture, loss, and training procedure, and differ only in whether a feature is treated as context. Across all eight datasets, CWAE improves the average AUCROC from 0.760 (WAE) to 0.797, with consistent gains on individual datasets such as bank (+0.033) and census (+0.067). This comparison directly captures the effect of conditional modeling, independent of how the context feature is chosen.
>
> Second, the effect of context choice is quantified through the CWAE Best analysis, which reports performance under the optimal context feature selected in hindsight. CWAE Best achieves an average AUCROC of 0.839 and ranks first on 6 out of 8 datasets, providing an upper bound on the gains achievable through improved context selection. The gap between CWAE and CWAE Best therefore isolates the additional benefit attributable specifically to selecting more informative contexts.
>
> Finally, our evaluation already enumerates all candidate context features for each dataset, which includes both highly informative and weak or uninformative choices. Empirically, poorly chosen contexts tend to regress toward the no-context baseline, while informative contexts yield substantial gains. While an explicit random-context ablation could be added, we believe it would largely mirror this observed behavior without providing additional insight beyond what is already captured by exhaustive context enumeration and the CWAE Best upper bound. Instead, we will clarify this decomposition more explicitly in the paper, as suggested.
>
> ---
>
> ## Requested Change 3
>
> >In Section 5.3 and Algorithm 1, clarify that the validation set is assumed to contain only normal samples, which is a standard but important assumption for unsupervised AD. This is mentioned later but should be explicit in the algorithm description.
>
> We will make this assumption explicit in both Section 5.3 and the description of Algorithm 1. In particular, we will clarify that the validation set is assumed to be predominantly normal, which is standard in unsupervised anomaly detection and underpins the reliability of validation-based context selection. By making this assumption explicit, this should improve clarity and avoid potential misinterpretation of the algorithm.

---

> > ### Comment · Reviewer_nSmt · 2026-01-21
> >
> > Thanks for the responses.

---

> ### Author Response · Authors · 2026-01-07
> **Author response to Reviewer nSmt 2/2**
>
> ## Broader Impact Concerns
>
> >The paper presents a methodological advancement in anomaly detection with primarily positive potential impacts (improved security, fraud detection). However, a Broader Impact Statement should be added to address the following:
>
> >i) Potential for Bias: The method's performance hinges on context selection. If the chosen context feature correlates with sensitive attributes (e.g., race, gender), the model could learn to flag anomalies differently for different demographic groups, potentially leading to discriminatory outcomes. The paper should briefly discuss this risk and the importance of auditing context features.
>
> >ii) Dual-Use Potential: Like all anomaly detection research, more effective tools could be used for both beneficial (network security) and harmful (surveillance) purposes. Acknowledging this duality is standard.
>
> >iii) Data and Reproducibility: The statement could highlight the use of publicly available benchmark datasets and the provided appendix as positive steps for reproducibility and ethical research practice. Currently, the absence of such a statement is a minor oversight.
>
> We agree and will add a dedicated Broader Impact Statement in the revised manuscript. This section will explicitly address (i) potential bias risks arising from context selection, including the possibility that context features correlated with sensitive attributes could lead to differential anomaly scoring across demographic groups and the importance of auditing context features in practice; (ii) the dual-use nature of anomaly detection methods, acknowledging both beneficial applications (e.g., security, fraud detection) and potential misuse (e.g., surveillance); and (iii) data transparency and reproducibility considerations, including the use of publicly available benchmark datasets and the supporting appendix, at a minimum. We will frame these points as considerations and limitations rather than claims of mitigation or guarantees.

---

### Review · Reviewer_ZRPQ · 2025-12-24

**Summary Of Contributions:**

Author proposes a Contextual Learning framework for unsupervised anomaly detection in tabular data. This framework change the modeling objective from the joint distribution P(X) to a conditional distribution P(Y|C), where C represents a context features and Y represents content features. Author argue that conditioning on a context features reduces variance and allows for sharper, context-specific decision boundaries compared to global models.

To implement this, the authors introduce the Conditional Wasserstein Autoencoder (CWAE), a deterministic reconstruction-based model. This method perform heuristic search to automatically select the optimal context feature from the available features. This strategy relies on minimizing the joint validation loss estimated after a single epoch of training. The method is evaluated on eight tabular datasets against six deep anomaly detection baselines and a standard WAE, demonstrating generally superior performance in terms of AUROC.

Strengths:
1. The authors evaluate their method against a wide range of baselines across datasets of varying complexity.
2. The theoretical motivation regarding the Law of Total Variance is sound and intuitive and proves need of contextual learning. Decomposing variance into within-context variance and between-context variance components provides a clear justification for why conditional modeling should theoretically outperform global modeling in heterogeneous data.


Weakeness:
1. While the authors state this is future work and justify the single-feature focus for interpretability, the framework's performance when dependencies are spread across multiple features i.e. P(Y|C_1,C_2) is not explored empirically.
2. The framework assume that the validation set is mostly normal. In truly unsupervised settings, ensuring a clean validation set is often impossible. The paper fails to investigate the failure mode where the validation set is contaminated.
3. The paper claims the model is effective for heterogeneous contexts. However, it does not provide evidence of how the model performs when the test set contains context values not present in the training set.
4. The paper discusses data complexity but ignores dimensional complexity. Training 38 models for 1 epoch takes ~18 minutes. If a user has a standard dataset with 500 features, this selection phase alone would take ~4 hours, whereas an Isolation Forest runs in seconds. This become much worse when dependencies are spread across multiple features

**Audience:**

Yes

**Audience Explanation:**

Anomaly detection in tabular data is a important problem in industry and a core topic in the machine learning community. The shift from global to contextual modeling is a relevant topic for researchers/practitioners dealing with high-cardinality categorical features that are common in real-world logs but often ignored or poorly modeled by standard anomaly detection algorithms.

**Claims And Evidence:**

Yes

**Claims Explanation:**

The authors provide both theoretical justifications and broad empirical evidence. The variance reduction section clearly explains why conditional modeling should theoretically improve detection sensitivity by reducing the noise of inter-context variability. Experimentally, author showed that CWAE outperforms its non-contextual counterpart and several state-of-the-art baselines across most datasets. The use of diverse datasets further supports the claim that the framework is generalizable.

**Requested Changes:**

1. Author should evaluate the model's performance on unseen contexts. perform a split where the Test Set contains only Context IDs that are not present in the Training Set. Author should discuss and empirically test how the model behaves for rare contexts (values of C with <10 samples). Does the model default to a prior, or does it overfit?
2. Author should quantify or discuss the cost of your selection algorithm relative to the number of features in the paper and compare with existing methods
3. Provide an experiment or discussion on the impact of anomaly contamination in the validation set. If the mostly normal assumption for the validation set is violated (e.g., 5-10% anomalies), does the algorithm still select the correct context feature?

---

> ### Author Response · Authors · 2026-01-07
> **Author response to Reviewer ZRPQ 1/2**
>
> Many thanks for the review and the helpful suggestions. We will respond to each requested change individually.
>
> ---
>
> ## Requested Change 1
>
> >Author should evaluate the model's performance on unseen contexts. Perform a split where the Test Set contains only Context IDs that are not present in the Training Set. Author should discuss and empirically test how the model behaves for rare contexts (values of C with <10 samples). Does the model default to a prior, or does it overfit?
>
> We agree that the behavior under unseen context values merits clarification. In the proposed framework, previously unseen context values are handled in a well-defined manner rather than causing model failure or uncontrolled extrapolation. When an unseen context is encountered at test time, it is mapped to a dedicated fallback representation in the context embedding space, causing the model to rely primarily on the learned content representation. This effectively regresses the detector toward a global (non-contextual) baseline in cases where context information is unavailable or uninformative.
>
> Importantly, our evaluation pipeline already distinguishes between seen and unseen contexts and allows detection behavior to be analyzed separately for these regimes, rather than conflating them in aggregate metrics. We will clarify this existing evaluation procedure and explicitly state the intended interpretation of results on unseen contexts in the revised manuscript.
>
> We emphasize that the goal of the current work is not to claim strong generalization to entirely novel contexts, but to show that contextual conditioning improves anomaly detection when meaningful context is present, while degrading gracefully when it is not. We will make this scope and design choice explicit.
>
> ---
>
> ## Requested Change 2
>
> >Author should quantify or discuss the cost of your selection algorithm relative to the number of features in the paper and compare with existing methods
>
> We appreciate this concern and will clarify the computational cost of the context selection procedure more explicitly in the manuscript. In the proposed approach, context selection requires training one epoch per candidate context feature rather than full convergence, resulting in a cost that scales linearly with the number of features. This procedure is trivially parallelizable across candidate contexts and is therefore well-suited to modern multi-core or distributed settings.
>
> We will also contextualize this cost relative to commonly used procedures in anomaly detection and representation learning, such as feature screening, hyperparameter search, and model selection via early stopping or proxy objectives, which similarly require multiple partial training runs. In particular, we will cite prior work that employs early-epoch training as a proxy for model quality and compare our selection strategy conceptually to these established practices. While our method incurs higher upfront cost than lightweight methods such as Isolation Forest, it remains practical for the target setting of high-cardinality, heterogeneous tabular data, where improved detection accuracy is often prioritized over minimal training time. We will make these trade-offs explicit in the revised manuscript.

---

> ### Author Response · Authors · 2026-01-07
> **Author response to Reviewer ZRPQ 2/2**
>
> ## Requested Change 3
>
> >Provide an experiment or discussion on the impact of anomaly contamination in the validation set. If the mostly normal assumption for the validation set is violated (e.g., 5-10% anomalies), does the algorithm still select the correct context feature?
>
> We agree that the composition of the validation set is an important consideration and will make the “mostly normal” assumption explicit earlier in the paper, both in Section 5.3 and in the description of Algorithm 1. This assumption is standard in unsupervised and semi-supervised anomaly detection, where both training and validation data are typically assumed to be predominantly normal. A large body of prior work adopts this setting, motivated by practical scenarios such as trusted historical baselines, clean initialization windows, or curated validation splits. Representative examples include one-class and deep SVDD–style methods ([Tax and Duin, 2004, Support Vector Data Description](https://link.springer.com/article/10.1023/B:MACH.0000008084.60811.49); [Ruff et al., 2018, Deep One-Class Classification](https://proceedings.mlr.press/v80/ruff18a/ruff18a.pdf)), as well as classification-based anomaly detection approaches trained solely on normal data ([Bergman and Hoshen, 2020, Classification-Based Anomaly Detection for General Data](https://arxiv.org/pdf/2005.02359)).
>
> We also note that robustness to significant anomaly contamination in unsupervised data constitutes a distinct and well-studied line of research, often framed under contaminated data, noisy supervision, or weakly supervised anomaly detection. Recent surveys explicitly identify this as a separate problem setting ([Jiang et al., 2023, Weakly Supervised Anomaly Detection: A Survey](https://arxiv.org/pdf/2302.04549)). Several methodological works propose dedicated mechanisms—such as iterative reweighting, pseudo-labeling, or auxiliary objectives—to address high contamination ratios, including Kim et al. ([2021, An Iterative Method for Unsupervised Robust Anomaly Detection under Data Contamination](https://arxiv.org/pdf/2309.09436)) and Wu et al. ([2022, Understanding and Mitigating Data Contamination in Deep Anomaly Detection: A Kernel-based Approach](https://www.ijcai.org/proceedings/2022/0322.pdf)). Our work addresses a different setting than these efforts and follows the conventional assumption that training and validation sets are mostly benign, which is shared by many widely used anomaly detection baselines.
>
> That said, our bilevel selection mechanism depends primarily on the relative ranking of candidate context features rather than absolute validation loss values. As a result, small amounts of contamination (e.g., 5–10%) are unlikely to fundamentally alter context rankings, consistent with prior observations that relative or early-learning signals often remain stable under limited noise. We will expand the discussion to clarify this point and to acknowledge that heavier contamination could degrade selection reliability, positioning contamination-robust extensions as a promising direction for future work.

---

### Author Response · Authors · 2026-01-01
**Acknowledgment of Reviewer Feedback**

We thank the reviewers for their careful reading of the manuscript and for their constructive and insightful feedback. We are encouraged that all reviewers found the paper’s claims to be supported by clear evidence, the theoretical motivation sound, and the problem of broad interest to the TMLR community. We acknowledge receipt of all reviews and are actively working to address the reviewers’ comments as promptly and thoroughly as possible.

---

### Author Response · Authors · 2026-02-23
**Thank You and Update on Camera-Ready Version**

We would like to thank the Action Editor and Reviewers for their time and effort in reviewing our submission. We sincerely appreciate the thoughtful comments, constructive feedback, and suggestions. We are grateful for the acceptance of our paper, and the reviews have been extremely helpful in improving the quality and clarity of our work.

We are currently working to incorporate the requested changes and to address all points raised in the reviews for the final camera-ready version. We are making good progress and expect to complete the revisions shortly.

Thank you again for your careful evaluation and guidance. We look forward to submitting the updated version soon.

---

### Decision · Action_Editor_2eeN · 2026-01-31

**Recommendation:** Accept with minor revision

**Additional Comments:**

Changes promised in the rebuttal have not been edited into the paper. I expect these changes to be edited in before final publishing.

**Audience:**

Yes

**Audience Explanation:**

The topic of anomaly detection in tabular data is clearly on scope. The novelty in the paper is relevant to many TMLR readers, as is agreed by all reviewers.

**Claims And Evidence:**

Yes

**Claims Explanation:**

All reviewers agree that the paper supports all claims being made. Minor initial concerns could be clarified in the rebuttal phase.